# ADAPTIVE CONTINUAL LEARNING: RAPID ADAPTATION AND KNOWLEDGE REFINEMENT

## ABSTRACT

Continual learning (CL) is an emerging research area aiming to emulate human learning throughout a lifetime. Most existing CL approaches primarily focus on mitigating catastrophic forgetting, a phenomenon where performance on old tasks declines while learning new ones. However, human learning involves not only retaining knowledge but also quickly recognizing the current environment, recalling related knowledge, and refining it for improved performance. In this work, we introduce a new problem setting, Adaptive CL, which captures these aspects in an online, possibly recurring task environment without explicit task boundaries or identities. We propose the LEARN algorithm to efficiently explore, recall, and refine knowledge in such environments. We provide theoretical guarantees from two perspectives: online prediction with tight regret bounds and asymptotic consistency of knowledge. Additionally, we present a scalable implementation that requires only first-order gradients for training deep learning models. Our experiments demonstrate that the LEARN algorithm is highly effective in exploring, recalling, and refining knowledge in adaptive CL environments, resulting in superior performance compared to competing methods.

## 1 INTRODUCTION

Inspired by the process of human lifelong learning, continuous learning (CL), also known as lifelong learning, aims to develop models that can sequentially learn tasks, simultaneously preserving and consolidating existing knowledge. The primary focus of CL approaches is on preventing catastrophic forgetting [Parisi et al., 2019, Van de Ven and Tolias, 2019], a phenomenon where the performance of previously learned tasks declines as new tasks are learned [McCloskey and Cohen, 1989]. Traditional CL literature [Kirkpatrick et al., 2017, Shin et al., 2017, Rebuffi et al., 2017, Lopez-Paz and Ranzato, 2017, Mallya and Lazebnik, 2018] mainly addresses a sequence of tasks with known task identities. In recent years, however, the focus has shifted towards more challenging scenarios in CL research, with growing interest in one scenario called task-free CL [Aljundi et al., 2019a, Lee et al., 2020, Jin et al., 2021, Pham et al., 2021, Ye and Bors, 2022a], where task identities and boundaries are unknown during training. In these instances, it becomes crucial for the learner to comprehend the current environment and incorporate new information without catastrophic forgetting, a challenging problem due to the lack of task identity.

Numerous approaches have been developed in CL to mitigate forgetting during the learning of new tasks. However, the situation involving potentially recurring tasks remains largely unexplored. This scenario presents additional challenges in both learning and employing, necessitating rapid task identification and recall of pertinent information to further enhance performance. This is a critical aspect of human lifelong learning. As humans encounter changing environments, they can swiftly recollect associated memories and adapt their learning when tasks switch, gradually building a knowledge base to improve their effectiveness and proficiency in recurring tasks. We believe such a learning process involves three key components: **quick recognition of new environments**, **recall of related knowledge**, and **refinement of existing knowledge**. Consequently, this highlights the need to explore the broader scope of CL problems beyond addressing catastrophic forgetting by incorporating all three above human-inspired capabilities.

In this paper, we introduce Adaptive CL, a new problem formulation designed to capture the multidimensional nature of human learning. This framework involves learning from dynamic online environments with possibly recurring tasks, while task boundaries and identities remain unknown.

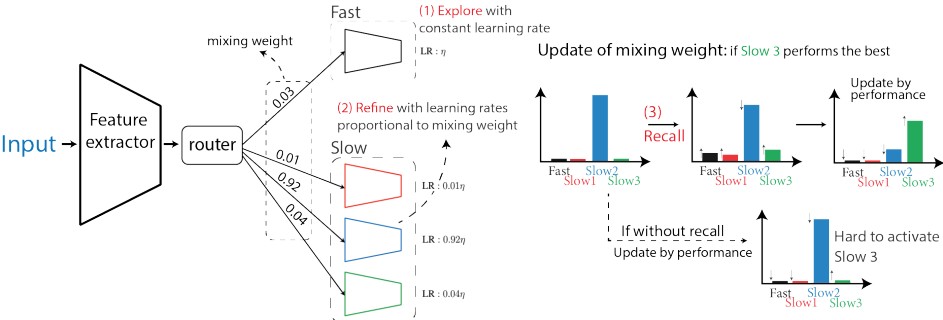

Figure 1: An illustration of our proposed, provably guaranteed LEARN algorithm with a fast learner, multiple slow learners, and a router keeping a mixing weight. After receiving the label, 1) **Exploration**: fast learner rapidly adapts to current data with a constant learning rate, while 2) **Refinement**: slow learners are refined with a learning rate proportional to the corresponding mixing weight. 3) **Recall:** The router mixes (recalls) mixing weight with slow model usage, then updates by performance. The recall step enables a rapid increase in the mixing weight of the best model.

To address the Adaptive CL problem, we will develop a novel approach that automatically unifies the above three decisions with provable performance guarantees. We propose LEARN (Lifelong Exploration, recAll, and Refinement of kNowledge), depicted in Figure 1, which adaptively integrates a fast learner with multiple slow learners through a router that keeps a mixing weight. The primary goal is to activate the relevant slow learner for improved performance on seen tasks, and to utilize the fast learner for identifying and quickly learning new tasks. During training, 1) **Exploration**: the fast learner greedily adapts to new data without memory, enhancing rapid learning of potential unseen tasks. 2) **Refinement**: slow learners are adaptively updated with learning rates proportional to mixing weight. 3) **Recall**: the mixing weight is first mixed with a slow weight, which records the usage of slow learners The recall process ensures the smallest value is above a certain value, making it faster to be pulled up to 1 if the corresponding slow learner behaves the best.

Furthermore, we establish a theoretical foundation for Adaptive CL to understand the learning limit from two perspectives: *performance-level* regret bound that ensures near-optimal decision-making, and *knowledge-level* consistency to the ground truth in hindsight. We provide theoretical guarantees that our developed approach is both effective in prediction and interpretable in knowledge acquisition in Adaptive CL. For application in large-scale models such as deep neural networks, we introduce a Gaussian mixture model (GMM) approximation of the proposed LEARN, which requires only first-order gradients for training deep learning models. This approximation enables efficient training while preserving the efficiency and interpretability of our approach for real-world applications. Our extensive experimental evaluation illustrates the success of the LEARN algorithm in several benchmark data cases.

## 1.1 MAIN CONTRIBUTIONS

• We propose Adaptive Continual Learning (Adaptive CL), a new framework inspired by human learning characteristics: rapid task recognition, efficient recall of related knowledge, and continuous refinement of knowledge. The framework aims to leverage the potential recurrence of tasks, e.g., math and history learning repeatedly throughout human life. The framework can be reduced to the conventional CL setup when there is no recurrence.

• To our best knowledge, we are the first to propose an algorithm, called LEARN, to take advantage of the potentially recurring nature of tasks, converting this challenge into an opportunity to improve performance over time. We establish a foundational theory regarding its performance in online prediction and interoperability in knowledge accumulation. In particular, we do not need to know the number of tasks, the recurrence patterns, or task identities when operating the LEARN algorithm. LEARN **adaptively** exploits recurrence whenever applicable to accelerate learning as time goes on. We also provide a scalable implementation of LEARN to facilitate its usage in deep learning.

• Through extensive experimental studies, we show the promising performance of the proposed LEARN algorithm. Our method notably improves accuracy over the online Finetune approach and closely aligns with the Oracle baseline that requires task identities. For instance, in CIFAR100, our method increases accuracy from $20.50\%$ to $43.26\%$, approaching the Oracle baseline of $45.50\%$.

Similar observations are made in other benchmark datasets, including CIFAR10, Mini-ImageNet, and Tiny-ImageNet. Furthermore, our method significantly outperforms the state-of-the-art online CL methods in the Adaptive CL context. For example, in CIFAR100, we improve from $30.47\%$, the best performance offered by existing methods, to $43.26\%$, which is a relative increase of $42\%$. Ablation studies show that removing any algorithmic component would negatively affect performance.

## 1.2 RELATED WORK

**Continual learning** Continual Learning (CL) targets learning in dynamic environments with restricted historical data access. Many existing works have achieved significant success in preventing catastrophic forgetting, namely preserving performance on old tasks while learning new tasks. Existing CL approaches can primarily be divided into three categories: regularization, replay, and dynamic architecture. Regularization-based methods [Kirkpatrick et al., 2017, Zenke et al., 2017, Lee et al., 2017, Li and Hoiem, 2017] minimize forgetting by imposing constraints on critical parameters from previous tasks. Replay approaches generate pseudo-samples [Shin et al., 2017] or store actual samples [Rebuffi et al., 2017, Rolnick et al., 2019] of prior tasks to implicitly protect essential parameters. Stored data can constrain optimization, preventing gradient updates in crucial directions [Lopez-Paz and Ranzato, 2017, Chaudhry et al., 2018a, Guo et al., 2020]. Finally, dynamic architecture methods either train separate masks of a dense neural network [Mallya and Lazebnik, 2018, Mallya et al., 2018, Serra et al., 2018] or maintain dynamic model structures [Rusu et al., 2016, Aljundi et al., 2017]. Experimental results demonstrate the superior performance of these methods in efficiently retaining knowledge and preventing catastrophic forgetting when training in a changing environment. Many recent works have theoretically investigated the cause of forgetting, specifically the impact of factors such as task similarity and ordering on generalization performance [Asanuma et al., 2021, Lin et al., 2023]. Additionally, the semi-supervised and unsupervised CL settings have also been studied [Yu et al., 2022, Achille et al., 2018].

**Task-free continual learning** Task-free CL presents a more complex scenario than traditional CL, as it aims to address unknown task boundaries and identities during training. In this setting, learners must retain knowledge to prevent catastrophic forgetting and quickly recognize the current task. Existing works have proposed replay-based and dynamic architecture methods. Replay-based methods [Jin et al., 2021, Aljundi et al., 2019b,c] maintain a small buffer of previous data and replay a small batch every step. The dynamic architecture approach expands the number of models by detecting a new task using the Dirichlet process [Lee et al., 2020] or discrepancy distances [Ye and Bors, 2022a,b].

In this paper, our objective is to improve the CL framework by integrating training and testing stages to better emulate realistic human learning scenarios. The learner must not only adapt to the changing environment, but also efficiently exploit knowledge by recalling and consolidating relevant information. This dual objective is analogous to the need for both exploration and exploitation in reinforcement learning [Kaelbling et al., 1996].

## 2 PROBLEM FORMULATION

Many recent works [Aljundi et al., 2019a, Lee et al., 2020, Jin et al., 2021, Ghunaim et al., 2023] have extended traditional Continual Learning (CL) to the task-free CL setting, where task boundaries and identities are unknown during training. However, these approaches are often evaluated without accounting for the potential task recurrence. Incorporating the possibility of recurring tasks in both the training and testing stages presents a more realistic but challenging framework. In this context, when the current task is recognized as having occurred previously, it is imperative to deploy and refine the related knowledge.

To better emulate human cognition, we introduce a novel problem setting, Adaptive CL, characterized by an online, recurring task environment without explicit task boundaries or identities. This setting presents challenges in rapidly recognizing, adapting, and refining knowledge in response to changes in task distribution. These abilities are essential for improved performance and realistic CL, closely resembling human learning capabilities.

We assume a sequential data stream $(x_t, y_t) \in \mathcal{X} \times \mathcal{Y}$ for time $t = 1, \ldots, T$. The learner is asked to predict the label $y_t$ with input $x_t$ based on historical data, $\{x_i, y_i\}_{i=1}^{t-1}$. In Adaptive CL, $(x_t, y_t)$ independently follows the unknown distribution $\mathcal{D}_t$, where the sequence of distributions $\mathcal{D}_1, \ldots, \mathcal{D}_T$ consists of $m_T$ distinct types of distributions and has $k_T - 1$ change points, namely

$$k_T \triangleq 1 + \sum_{t=1}^{T-1} \mathbb{1}(\mathcal{D}_{t+1} \neq \mathcal{D}_t) < T, \quad m_T \triangleq \mathrm{Card}(\{\mathcal{D}_t\}_{t=1}^T) < k_T,$$

where $\text{Card}(\cdot)$ denotes the set cardinality. For simplicity, we omit the subscript $T$ in $m_T$ and $k_T$, and assume that the $m$ **modes**, namely distinct distributions, are $\{\tilde{\mathcal{D}}_j\}_{j=1}^m$. The Adaptive CL objective is twofold: 1) to achieve a small cumulative loss by enabling the learner to swiftly adapt to previously learned tasks, and 2) to allow the learned knowledge to converge to the underlying ground truth. To better quantify these objectives, we propose two key questions:

**Question 2.1** (Regret Bound). Given a model class $\mathcal{M} \triangleq \{M(\,\cdot\,;\theta) : \mathcal{X} \mapsto \mathcal{Y}, \theta \in \Theta\}$, what is the *optimal upper bound* for the cumulative expected regret with respect to the best competitors from hindsight? The cumulative expected regret for a randomized algorithm $\mathcal{A}$ is defined as

$$\mathbb{E}\left[\text{Regret}_T\right] \triangleq \sum_{t=1}^T \mathbb{E}\left[l_t^{\mathcal{A}}\right] - \sum_{t=1}^T \min_{\theta \in \Theta} \mathbb{E}\left[l_t(\theta)\right],$$

where $l_t(\theta) \triangleq L\left(M(x_t;\theta), y_t\right)$ for a given loss function $L$, and $l_t^{\mathcal{A}} \triangleq \mathbb{E}_{\theta \sim \pi_t(\mathcal{A})}[l_t(\theta)]$ with $\pi_t(\mathcal{A})$, a distribution over $\Theta$ produced by algorithm $\mathcal{A}$.

A small regret bound in Question 2.1 implicitly guarantees the swift adaptation to previously learned tasks, which is essential for performance improvement. Otherwise, the learner will learn from scratch for every recurring task, which is suboptimal.

**Question 2.2** (Knowledge Convergence). Will knowledge converge to the ground truth?

One critical aspect of human learning underexplored in the CL literature, to the best of our knowledge, is how the learner could swiftly adapt to previously learned tasks. This aspect is implicitly guaranteed by the regret bound formulated in Question 2.1. In addition, the understanding of knowledge refinement has been limited in the literature. Our work, through Question 2.2, aims to contribute to a deeper understanding of knowledge convergence and its role in Adaptive CL.

## 3 PROPOSED ADAPTIVE CL SOLUTION: LEARN ALGORITHM

In this section, we develop LEARN (Lifelong Exploration, recAll, and Refinement of kNowledge), a novel algorithm designed to address the challenges of adaptive CL through exploration, knowledge recall, and refinement. This approach facilitates swift adaptation and ongoing consolidation of knowledge. The intuition and detailed explanation of LEARN can be found in Section 3.1, while theoretical guarantees demonstrating its effectiveness are provided in Section 3.2. In Section 3.3, we discuss the scalable implementation of LEARN, highlighting its applicability in deep learning.

### 3.1 ALGORITHM DESCRIPTION

The LEARN algorithm, as shown in Algorithm 1, consists of two main components: fast and slow learners. Fast learner absorbs new data in the adaptive CL environment through exploration, using tempered Bayesian updates [Erven et al., 2011, Kirkpatrick et al., 1983, Friel and Pettitt, 2008]. The slow learner consolidates previously learned information, laying the foundation for swift recognition and adaptation to recurring tasks for later knowledge recall. LEARN operates in three steps: 1) *exploration*, 2) *recall*, and 3) *refinement*. During exploration, the algorithm processes new data with the fast learner. In the recall stage, the fast learner recalls stored knowledge by mixing with the slow learner, facilitating rapid adaptation to previously learned tasks. Finally, in the refinement stage, the information from the fast learner is integrated into the slow learner using a mixing process to enhance the quality of the stored information. As illustrated in Figure 2, when the task switches to a previously learned task, a direct update without recall attempts to increase a small mass around the optimal parameter. However, once mixed with the slow density, that is, recall, the mass around the optimal parameter increases, resulting in faster recognition of the learned task.

When receiving the input $x_t$ at time $t$, the agent randomly samples $\hat{\theta}_t$ from the fast learner $f_{t-1}$ and provides prediction $\hat{y}_t = M(x_t, \hat{\theta}_t)$. Upon receiving the true label $y_t$, the fast learner is updated in the exploration stage, Line 5, using the tempered Bayesian update with temperature $\eta$. The introduction of the temperature parameter $\eta$ is designed to moderate the impact of varying loss scales, especially since our approach does not rely on a probabilistic setting or the use of negative log-likelihood loss.

When learning from stationary data, this update leads the fast learner to converge exponentially fast to the point mass on the minimizer. However, it is an undesirable feature in non-stationary environments due to the long time required to increase the exponentially small probability on the new minimizer. To address this, in the recall stage, the fast learner recalls the slow learner $g_{t-1}$ with a recall ratio

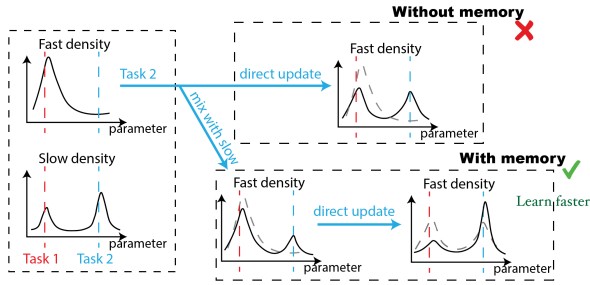

Figure 2: Comparison between fast density updates without and with memory during transition from Task 1 to Task 2. A memoryless direct (Bayesian) update begins with minimal mass around optimal parameter of Task 2, while fast density after recall starts the update with a larger mass.

$\alpha_t \in [0, 1]$ (Line 6). This mixing step enables the agent to quickly adapt to previously learned tasks, as the mass of $g_{t-1}$ near the corresponding minimizer is relatively large. In the refinement stage (Line 7), the slow learner $g_{t-1}$ is consolidated with the fast learner $\tilde{f}_t$ using a learning rate $\gamma_t \in [0, 1]$.

---

**Algorithm 1** LEARN: Lifelong Exploration, recAll, and Refinement of kNowledge

**Input** Model class $\{M(\cdot; \theta) : \theta \in \Theta\}$, mixing $\{\alpha_t \in [0, 1]\}_{t=1}^T$, forgetting $\{\gamma_t \in [0, 1]\}_{t=1}^T$, temperature $\eta > 0$ .
1: Initialization: $f_0(\theta) = g_0(\theta) = 1/\text{Vol}(\Theta)$.
2: **for** $t = 1 \to T$ **do**
3:      Receive $x_t$, randomly sample $\hat{\theta}_t$ from density $f_{t-1}$, and predict $\hat{y}_t = M(x_t; \hat{\theta}_t)$.
4:      Receive $y_t$ and the corresponding loss $l_t(\theta) \triangleq L(M(x_t; \theta), y_t)$.
5:      Exploration: $\tilde{f}_t(\theta) \leftarrow f_{t-1}(\theta) \exp\{-\eta l_t(\theta)\}$, and normalize $\tilde{f}_t$.
6:      Recall: $f_t(\theta) \leftarrow (1 - \alpha_t)\tilde{f}_t(\theta) + \alpha_t g_{t-1}(\theta)$.
7:      Refinement: $g_t(\theta) \leftarrow (1 - \gamma_t)g_{t-1}(\theta) + \gamma_t \tilde{f}_t(\theta)$.

---

The exploration, recall, and refinement stages of LEARN collectively promote rapid adaptation and improved performance in previously learned tasks. To further elucidate the adaptability and knowledge convergence, we will delve into its theoretical underpinnings in subsequent analysis.

### 3.2 THEORETICAL ANALYSIS AND INSIGHTS INTO THE ADAPTIVENESS

In our theoretical analysis, we first tackle Question 2.1 by providing a regret upper bound for Algorithm 1 in Proposition 3.1. The technical details are included in the Appendix.

**Proposition 3.1.** *Assume set $\Theta \subseteq \mathbb{R}^d$ is compact with $\sup_{\theta \in \Theta} \|\theta\|_2 \leq D$, and $|l_t(\theta) - l_t(\theta')| \leq Z_t \|\theta - \theta'\|_2$ for all $\theta, \theta' \in \Theta$, with $\mathbb{E}[Z_t^2] \leq v^2$ . Then there exists $\eta_{opt} > 0$, stated in the Appendix, such that Algorithm 1 with $\alpha_t = k/T$ and $\gamma_t = 1/t$ yields an expected cumulative regret*

$$\begin{aligned}
\mathbb{E}\left[Regret_T\right] \leq & Dv\sqrt{2T\left(md\log\frac{DvT}{2} + 2k\log\frac{T}{k} + k\log mk + md\right) + 1} \\
= & O\left(Dv\sqrt{T}\sqrt{md\log\{DvT\} + k\log\{mT\}}\right).
\end{aligned} \tag{1}$$

**Leveraging adaptation to reduce dimensionality costs.** In the upper bound, we observe two distinct sources of loss. Excluding the shared $Dv\sqrt{T}$, the first term, $\sqrt{md\log\{DvT\}}$, is dimension-related and signifies the cost of learning a new distribution, occurring $m$ times. This dimension-related aspect is particularly crucial in deep learning, where large dimensions are commonplace. The fact that it is not related to $k$ indirectly demonstrates the rapid adaptation and knowledge refinement capabilities of LEARN. The second term, $\sqrt{k\log mT}$, is dimension-free and encapsulates the information needed to identify task boundaries and the cost associated with retaining the current distribution. This component has also been recognized and explored in the expert learning literature [Koolen et al., 2012, Robinson and Herbster, 2021].

Next, we turn our attention to the knowledge refinement, specifically the convergence addressed in Question 2.2, by presenting Proposition 3.2. This crucial result demonstrates that the knowledge in

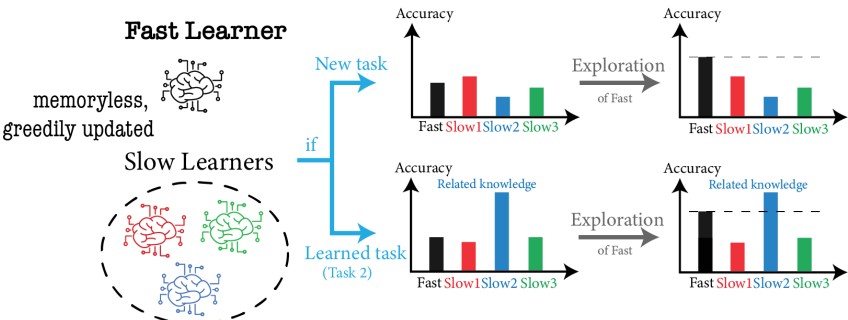

Figure 3: Comparison of fast and slow learners: A fast learner quickly excels in **unseen** tasks post-exploration, outperforming slow learners. However, for **seen** tasks, one relevant slow learner, integrating past data, surpasses fast learner even after exploration.

Algorithm 1 indeed attains the desired convergence properties. We denote the frequency of mode $\tilde{\mathcal{D}}_j$ as $\text{freq}_{T,j} \triangleq \sum_{t=1}^T \mathbb{1}(\mathcal{D}_t = \tilde{\mathcal{D}}_j)/T$ for any $n \in \mathbb{N}$.

**Proposition 3.2** (Convergence). *Under the assumptions in Proposition 3.1. Suppose $\{\mathbb{E}[l_t]\}_{t=1}^T$ is uniformly strict, namely for any $\varepsilon > 0$, there exists $\delta > 0$ such that,*

$$\min_{1 \leq t \leq T} \inf_{\theta:d(\theta,\mathcal{C}_t) \geq \varepsilon} \{\mathbb{E}[l_t(\theta)] - \min_{\theta'} \mathbb{E}[l_t(\theta')]\} \geq \delta,$$

*where the set of minimizers $\mathcal{C}_t \triangleq \arg\min_{\theta \in \Theta} \mathbb{E}[l_t(\theta)]$. If $k = o(T/\log T)$, then there exists $\eta_{opt,T}$ such that Algorithm 1 with $\alpha_t = k/T$ and $\gamma_t = 1/t$ has the following properties:*

1. *For any $\varepsilon > 0$, $\lim_{T \to \infty} \mathbb{E} \int_{\theta \in \Theta:d(\theta, \cup_t \mathcal{C}_t) \geq \varepsilon} g_T(\theta) = 0$.*

2. *If further assume for $\lim_{T \to \infty} \text{freq}_{T,j} = q_j$ and the minimizer $\{\mathcal{B}_j\}_{j=1}^m$ are disjoint. Then*

$$\lim_{\varepsilon \to 0} \lim_{T \to \infty} \mathbb{E} \int_{\theta \in \Theta:d(\theta,\mathcal{B}_j) \leq \varepsilon} g_T(\theta) = q_j,$$

*where $\mathcal{B}_j \triangleq \arg\min_{\theta \in \Theta} \mathbb{E}_{(x,y) \sim \tilde{\mathcal{D}}_j} [L(M(x;\theta), y)]$.*

**From black-box to knowledge building.** The convergence result presented in Proposition 3.2 provides a mathematical insight into knowledge building, unlike many existing black-box heuristic CL approaches. Our analysis illuminates the core mechanisms that underpin the adaptive capabilities of the LEARN algorithm, fostering a comprehensive understanding of its inner workings.

In summary, Propositions 3.1 and 3.2 address Questions 2.1 and 2.2, respectively. LEARN effectively adapts to previously learned tasks and refines its knowledge base, exhibiting key aspects of human learning and making it suitable for various real-world applications. However, Algorithm 1 may not be scalable for large-scale deep learning tasks due to its requirement for density integration. To tackle this, the following subsection introduces an approximation of LEARN that employs an efficient approximation method, enhancing its scalability and compatibility with deep learning applications, and thereby extending its applicability.

### 3.3 SCALABLE IMPLEMENTATION

In the previous subsection, we introduced LEARN in Algorithm 1 and provided theoretical guarantees. However, this approach, which requires density over a high dimensional space, faces scalability challenges in large-scale deep learning tasks. To address this issue, we present Scalable LEARN in Algorithm 2, an efficient approximation using Gaussian Mixture Models (GMMs). While Variational Inference (VI) [Jordan et al., 1999] is a popular technique for approximating target distributions in deep learning literature, it is not well-suited for our problem setting due to the recursive form in Algorithm 1. GMM, on the other hand, offers a more straightforward and effective solution while preserving the core properties and adaptability of the LEARN algorithm.

Intuitively, the slow density is a mixture of components, with each component concentrating around the optimal parameter of a previously seen task. Consequently, we approximate densities in Algorithm 1 using a Gaussian Mixture Model (GMM):

$$\tilde{f}_t(\theta) \approx w_{t,0}\mathcal{N}(\theta_t, \sigma^2 I_d) + \sum_{i=1}^{\hat{m}_t} w_{t,i}\mathcal{N}(\beta_{t,i}, \sigma^2 I_d),$$

$$g_t(\theta) \approx r_{t,0}\mathcal{N}(\theta_t, \sigma^2 I_d) + \sum_{i=1}^{\hat{m}_t} r_{t,i}\mathcal{N}(\beta_{t,i}, \sigma^2 I_d),$$

where $\theta_t$ represents the fast learner, and $\beta_{t,i}$ for each $i \in [\hat{m}_t]$ corresponds to the slow learners. By substituting this approximation into the Algorithm 1 and applying Taylor's expansion, the original density updates are transformed into first-order parameter updates in Algorithm 2. The detailed technical derivation is included in Appendix B.2.

At time $t + 1$, fast and slow learners hold 1 and $\hat{m}_t$ models, respectively. The target prediction is computed as the adaptively weighted average of the predictions from these $\hat{m}_t + 1$ models, with the combining weights $w_t$ updated according to Line 6 of Algorithm 2. The fast learner, updating only on the most recent data, operates without memory retention. Following this, the models in the slow learner are updated with a learning rate adaptive to the combining weights. As illustrated in Figure 3, when introduced to a new task, the fast learner quickly adapts and outperforms models in the slow learner. However, when dealing with previously encountered tasks, the affiliated model in the slow learner surpasses the memoryless fast learner. Such different behaviors toward seen and unseen tasks enable task identification through updates in Line 6. The patience tracks the high value of the fast learner combining weight since a model was last added to the slow learner. If patience crosses a set threshold, indicating a new task, a snapshot of the fast learner is directly added to the slow learner.

---

**Algorithm 2** Scalable LEARN

**Input** Model class $\{M(\cdot; \theta) : \theta \in \Theta\}$, mixing $\{\alpha_t\}_{t=1}^T$, temperature $\eta > 0$, variance $\sigma^2$, patience $Q > 0$.

**Output** Fast learner $\theta_T$, slow learners $\beta_{T,i}$ for $i \in [\hat{m}_T]$

1: Initialization: fast learner $\theta_0 \sim \text{Unif}(\Theta)$, predictive weight $w_{0,0} = 1$, cache weight $r_{0,0} = 1$, slow learner $\mathcal{G}_0 = \emptyset$, patience $q_0 = 0$.

2: **for** $t = 1 \to T$ **do**

3:    Receive $x_t$ and predict $\hat{y}_t = w_{t-1,0}M(x_t; \theta_{t-1}) + \sum_{i=1}^{\hat{m}_{t-1}} w_{t-1,i}M(x_t; \beta_{t-1,i})$.

4:    Receive $y_t$ and corresponding loss $l_t(\theta)$.

5:    Exploration of fast learner: $\theta_t \leftarrow \theta_{t-1} - \eta\sigma^2 \nabla l_t(\theta_{t-1})$.

6:    Knowledge recall for adaptation: ($i \in [\hat{m}_{t-1}], \beta_{t-1,0} \triangleq \theta_{t-1}$)
$$w_{t,i} \leftarrow \{(1-\alpha)w_{t-1,i} + \alpha r_{t-1,i}\} \exp\{-\eta l_t(\beta_{t-1,i})\},$$

7:    Refinement of knowledge: for $i \in [\hat{m}_{t-1}]$
$$r_{t,i} \leftarrow r_{t-1,i} - \frac{1}{t}(r_{t-1,i} - w_{t,i}), \quad \beta_{t,i} \leftarrow \beta_{t-1,i} - \eta\sigma^2 \frac{w_{t,i}}{tr_{t,i}} \nabla l_t(\beta_{t-1,i}).$$

8:    Update patience: $q_t \leftarrow q_{t-1} + \max\{0, w_{t,0} - 1 + \alpha\}$

9:    **if** patience $q_t > Q$ **then**

10:        Consolidate knowledge with cache, and initialize cache:
$$\mathcal{G}_t \leftarrow \mathcal{G}_{t-1} \cup \{(r_{t,0}, \theta_t)\}. \quad q_t = r_{t,0} = 0$$

---

## 4 EXPERIMENTAL EVALUATION

### 4.1 ADAPTIVE CL DATASTREAM

We conduct extensive experiments to evaluate the performance, the ability to adapt to learned tasks, and the knowledge quality. Recall that an Adaptive CL scenario consists of a data stream from an online, non-stationary environment with potentially recurring tasks and unknown task boundaries or identities. To emphasize the challenge of the problem, we create multiple tasks with distinct original labels, which are then re-labeled within the same label region–otherwise, the task boundaries and identity could be inferred directly from the labels. In the following experiments, each data point is presented only once with batch size 5.

**Datasets. CIFAR10** [Krizhevsky et al., 2009] consists of color images in 10 classes, with 6000 images per class. We create 5 tasks from CIFAR10 by splitting the dataset into 5 subsets according to labels $(0/1, 2/3, \ldots, 8/9)$. Each task is randomly split into 40 segments with 50 batches per segment. By shuffling and combining all 200 segments, we obtain the **Adaptive CIFAR10** scenario.

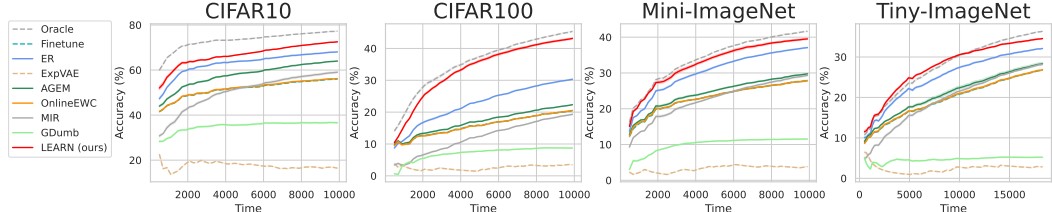

Figure 4: The running average accuracy of all compared methods on the three Adaptive scenarios from 10 runs. The dashed line indicates non-Adaptive CL methods with task identity information.

Table 2: Comparison of Average Accuracy (%) (mean ± se) from 10 runs. * represents non-task-free baselines.

| Method | CIFAR10 | CIFAR100 | Mini-ImageNet | Tiny-ImageNet |
|---|---|---|---|---|
| Oracle* | $77.38 \pm 0.08$ | $45.50 \pm 0.13$ | $41.68 \pm 0.10$ | $36.37 \pm 0.07$ |
| ExpVAE* | $16.73 \pm 0.03$ | $3.45 \pm 0.02$ | $3.80 \pm 0.02$ | $2.82 \pm 0.02$ |
| Finetune | $56.12 \pm 0.09$ | $20.50 \pm 0.12$ | $27.93 \pm 0.18$ | $26.80 \pm 0.12$ |
| ER | $68.19 \pm 0.22$ | $30.47 \pm 0.14$ | $37.13 \pm 0.15$ | $32.12 \pm 0.10$ |
| A-GEM | $64.16 \pm 0.11$ | $22.40 \pm 0.07$ | $29.81 \pm 0.14$ | $28.40 \pm 0.35$ |
| Online EWC | $56.36 \pm 0.10$ | $20.58 \pm 0.03$ | $27.86 \pm 0.14$ | $26.88 \pm 0.15$ |
| MIR | $59.22 \pm 0.40$ | $19.39 \pm 0.18$ | $29.41 \pm 0.22$ | $28.28 \pm 0.12$ |
| GDumb | $36.69 \pm 0.24$ | $8.74 \pm 0.06$ | $11.50 \pm 0.03$ | $5.22 \pm 0.07$ |
| LEARN | $\mathbf{72.70 \pm 0.07}$ | $\mathbf{43.26 \pm 0.25}$ | $\mathbf{39.54 \pm 0.19}$ | $\mathbf{34.57 \pm 0.12}$ |

**CIFAR100** [Krizhevsky et al., 2009] consists of color images in 100 classes, each with 600 images. Like before, we create 10 tasks from CIFAR100 by splitting it according to labels, so there are 10 classes per task. We then obtain the **Adaptive CIFAR100** scenario with 20 segments per task, in a way similar to CIFAR10. **Mini-ImageNet** and **Tiny-ImageNet** [Le and Yang, 2015] contains 100 and 200 classes respectively, and we obtain the **Adaptive Mini-ImageNet** and **Adaptive Tiny-ImageNet** scenarios similar to the Adaptive CIFAR100 with 10 tasks and 20 segments per task.

**Compared methods.** Except for our method, LEARN, we further evaluate the following methods, where * indicates unrealistic baselines: 1) **Finetune** with a neural network naively trained on the new data. 2) **Oracle*** as the upper performance limit consisting of multiple models, where one corresponds to one task, with known task identities during training and testing. 3) **ExpVAE*** (Expansion+VAE) consisting of (classifier, generator) tuples, which is a popular structure in dynamic expansion with mixture models in Task-free CL [Lee et al., 2020, Ye and Bors, 2022a,b]. We assume that the task identity is known

Table 1: Number of trainable parameters (in million) for 4 datasets. * represents non-task-free baselines.

| Method | CIFAR10 | CIFAR100 | Mini. | Tiny. |
|---|---|---|---|---|
| Oracle* | 11.19 | 11.68 | 12.73 | 13.24 |
| ExpVAE* | 12.33 | 12.81 | 20.06 | 20.58 |
| LEARN | 11.22 | 12.19 | 13.24 | 14.27 |
| Others | 11.17 | 11.22 | 12.27 | 12.32 |

during training. However, during the prediction stage, the task identity must be inferred by the generators, Variational Autoencoder (VAE) [Kingma and Welling, 2013]. 4) **ER** (Experience Replay) [Chaudhry et al., 2019] 5) **A-GEM** (Averaged Gradient Episodic Memory) [Chaudhry et al., 2018a] 6) **Online EWC** [Chaudhry et al., 2018b] 7) **MIR** [Aljundi et al., 2019d] 8) **GDumb** [Prabhu et al., 2020]. The details of methods 4)-8) are included in Appendix D.1.

**Architecture.** We employ ResNet18 [He et al., 2016] as the backbone for all methods. Methods like Oracle and LEARN, which utilize multiple models, share this same backbone but feature multiple output layers, each consisting of two fully connected linear layers. As indicated in Table 1, the total parameter count is comparable across all methods, except ExpVAE. This method incorporates multiple VAEs, utilizing two convolutional layers in both the encoder and decoder. The learning rate is set to 0.001 with the SGD optimizer.

**Metrics.** We consider three metrics: 1) **Average Accuracy**: the cumulative accuracy divided by the total time: $\sum_{t=1}^{T} \text{Acc}_t / T$, where $\text{Acc}_t$ is the accuracy at time $t$. 2) **Knowledge Accuracy**: the mean of test accuracy over all tasks: $\sum_{i=1}^{m} \text{TestAcc}_i / m$, where $\text{TestAcc}_i$ is the test accuracy of $i$-th task giving task identity. 3) **Adaptiveness**: the weighted average of accuracy, where the weights decay geometrically with factor $\gamma \in [0, 1]$ and reinitialize whenever the task changes, detailed in Appendix

Table 3: Comparisons of Knowledge Accuracy (%) (mean ± se) from 10 runs. * represents non-task-free baselines.

| Method | CIFAR10 | CIFAR100 | Mini-ImageNet | Tiny-ImageNet |
|---|---|---|---|---|
| Oracle* | 81.91 ± 2.64 | 51.92 ± 1.14 | 50.85 ± 1.43 | 43.08 ± 1.11 |
| ExpVAE* | 85.17 ± 3.00 | 47.74 ± 1.27 | 42.35 ± 1.54 | 33.27 ± 1.09 |
| Finetune | 18.25 ± 8.75 | 8.91 ± 4.04 | 5.72 ± 2.12 | 4.48 ± 4.07 |
| ER | 43.48 ± 6.71 | 18.64 ± 3.90 | 18.67 ± 3.47 | 8.01 ± 3.15 |
| A-GEM | 25.45 ± 8.11 | 10.83 ± 4.55 | 6.90 ± 2.43 | 4.71 ± 3.96 |
| Online EWC | 18.25 ± 8.75 | 9.39 ± 2.21 | 5.59 ± 2.15 | 4.31 ± 3.93 |
| MIR | 18.49 ± 8.84 | 10.39 ± 3.85 | 7.80 ± 2.38 | 5.36 ± 3.92 |
| GDumb | 37.71 ± 4.15 | 9.66 ± 1.01 | 12.66 ± 0.98 | 5.42 ± 0.53 |
| LEARN | **75.04 ± 5.10** | **41.22 ± 2.02** | **36.08 ± 4.98** | **36.98 ± 1.97** |

Table 4: Comparison of Adaptiveness (mean±se) with $\gamma = 0.99$ from 10 runs in $10^{-2}$ scale. * represents non-task-free baselines.

| Method | CIFAR10 | CIFAR100 | Mini-ImageNet | Tiny-ImageNet |
|---|---|---|---|---|
| Oracle* | 77.31 ± 0.09 | 45.37 ± 0.12 | 41.54 ± 0.11 | 35.65 ± 0.07 |
| ExpVAE* | 16.67 ± 0.02 | 3.37 ± 0.02 | 3.92 ± 0.02 | 2.92 ± 0.03 |
| Finetune | 53.32 ± 0.09 | 18.37 ± 0.10 | 25.74 ± 0.17 | 21.36 ± 0.10 |
| ER | 67.00 ± 0.23 | 28.55 ± 0.13 | 35.96 ± 0.14 | 28.66 ± 0.10 |
| A-GEM | 61.94 ± 0.12 | 20.18 ± 0.07 | 27.68 ± 0.12 | 22.95 ± 0.31 |
| Online EWC | 53.57 ± 0.10 | 18.42 ± 0.03 | 25.70 ± 0.13 | 21.40 ± 0.14 |
| MIR | 56.79 ± 0.43 | 17.78 ± 0.18 | 27.45 ± 0.21 | 23.27 ± 0.08 |
| GDumb | 36.75 ± 0.24 | 8.81 ± 0.07 | 11.44 ± 0.03 | 5.19 ± 0.06 |
| LEARN | **71.85 ± 0.08** | **42.23 ± 0.27** | **38.68 ± 0.18** | **32.82 ± 0.15** |

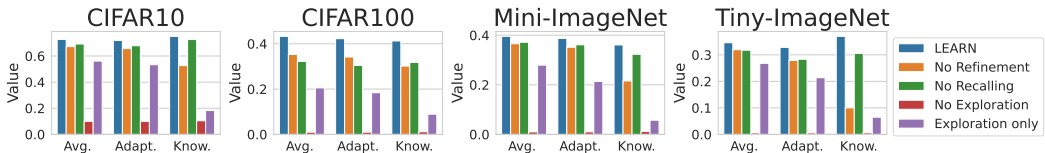

Figure 5: Ablation study of the impact removing any key component of LEARN on CIFAR10, CIFAR100, Mini-ImageNet, and Tiny-ImageNet over 5 runs in terms of Average Accuracy (Avg.), Adaptiveness (Adapt.) and Knowledge Accuracy (Know.).

D.4: $\frac{\sum_{i=1}^{k} \sum_{\tau=1}^{n_{i+1}-n_i} \gamma^{\tau-1} \mathrm{acc}_{n_i+\tau}}{\sum_{i=1}^{k} \sum_{\tau=1}^{n_{i+1}-n_i} \gamma^{\tau-1}}$, where $n_0 < \ldots < n_k$ are segment boundaries. Larger adaptiveness means a faster speed of recall of related information.

**Results.** As illustrated in Figure 4 and Table 2, the average accuracy of LEARN markedly surpasses that of competing methods across all scenarios. Among the methods, ER is a simple but strong comparator. Although ExpVAE utilizes the information of task identities in training, the dependency on VAE to recognize current task in the test stage significantly harms the performance due to the complexity of training VAE.

Table 3 measures the mean test accuracy in all tasks, representing the quality of knowledge refinement. The refined knowledge in LEARN, namely a mixture of models, is significantly better than competing methods. Table 4 measures the Adaptiveness, defined in Metrics,

As shown in the table, LEARN has the largest adaptiveness, showing the ability to adapt to learned tasks more efficiently. It is worth noting that the competing CL approaches in the literature were not designed and optimized for the Adaptive CL scenario, leading to less satisfactory performance. More discussions of the experimental results are included in the Appendix.

### 4.2 ABLATION STUDIES

**Impact of key stages in LEARN** We ablate the three core stages of LEARN, Algorithm 2, namely, exploration, recall and refinement of knowledge on 4 datasets, presented in Figure 5. Removing any component of LEARN adversely affects performance. In particular, the absence of exploration leads to a significant performance drop. This is because, without exploration, the algorithm is unable to detect new tasks by comparing knowledge to the fast learner.

**Impact of hyperparameters** We evaluate a variety of patience thresholds $Q$ from 1 to 20, and mixing parameters $\alpha$ from 0.1 to 0.5. The details are included in the Appendix. The choice of patience threshold and mixing parameter exhibit minimal influence on the average accuracy and adaptiveness. of knowledge.

### 5 CONCLUSION

In this work, we proposed a realistic and challenging problem, Adaptive CL, and two key characteristics: performance and knowledge quality. To address the problem, we propose a unified LEARN algorithm that simultaneously recognizes the current task, recalls related information, and refines knowledge. Theoretical analysis of LEARN guarantees near-optimal performance and asymptotically consistent knowledge. To be efficient in deep learning, we propose a scalable implementation. The experimental results show that LEARN significantly surpasses the baseline methods in multiple aspects. The **Appendix** contains additional details on implementation details, more ablation studies, and technical proofs. We do not envision any negative social impact of this work.

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

# A    THEORETICAL ANALYSIS

## A.1    PROOF OF PROPOSITION 3.1

*Proof.* We first show useful lemmas as follows. Lemma A.2 upper bounds the regret by the potential in the form $-\log\{\mathbb{E}[\exp\{-\eta \cdot \text{regret}\}]\}/\eta$. Lemma A.3 captures the local continuity utilizing the Lipchitz-like condition $|l_t(\theta) - l_t(\theta')| \leq Z_t\|\theta - \theta'\|_2$.

**Lemma A.1** (Hoeffding's lemma). *Let $X$ be a bounded random variable with $X \in [a, b]$. Then for any $\lambda \in \mathbb{R}$,*

$$\mathbb{E}[\exp\{\lambda(X - \mathbb{E}[X])\}] \leq \exp\{\frac{\lambda^2(b-a)^2}{8}\}$$

**Lemma A.2.** *If random variable $h(\theta)$ satisfies $|h(\theta) - h(\theta')| \leq Z\|\theta - \theta'\|_2$ for all $\theta, \theta' \in \Theta$ with $\sup_{\theta \in \Theta}\|\theta\| \leq D$, then for $\theta^* \in \Theta$, any density $f(\theta)$ over $\Theta$ and $\eta > 0$,*

$$\int_\Theta f(\theta)\{h(\theta) - h(\theta^*)\}d\theta \leq -\frac{1}{\eta}\log\int_\Theta f(\theta)\exp(-\eta\{h(\theta) - h(\theta^*)\}) + \frac{\eta D^2 Z^2}{2}$$

*Proof.* For any $\theta \in \Theta$, by Hoeffding's lemma on $h(\tilde{\theta}) - h(\theta^*)$ where $\tilde{\theta} \sim f$ with $\lambda = -\eta$, since $\|h(\tilde{\theta}) - h(\theta^*)\| \leq Z\|\theta - \theta^*\| \leq 2ZD$, we have

$$\mathbb{E}\left[\exp\left\{-\eta\left(h(\tilde{\theta}) - h(\theta^*) - \mathbb{E}\left[h(\tilde{\theta}) - h(\theta^*)\right]\right)\right\}\right] \leq \exp\{\frac{\eta^2 D^2 Z^2}{2}\}.$$

Therefore,

$$\mathbb{E}\left[\exp\left\{-\eta\left(h(\tilde{\theta}) - h(\theta^*)\right)\right\}\right]\exp\left\{\eta\mathbb{E}\left[h(\tilde{\theta}) - h(\theta^*)\right]\right\} \leq \exp\{\frac{\eta^2 D^2 Z^2}{2}\}.$$

Taking log on both sides, we have

$$\int_\Theta f(\theta)\{h(\theta) - h(\theta^*)\}d\theta \leq -\frac{1}{\eta}\log\int_\Theta f(\theta)\exp\{-\eta\{h(\theta) - h(\theta^*)\}\}d\theta + \frac{\eta D^2 Z^2}{2}.$$

$\square$

**Lemma A.3.** *Under the assumptions of Lemma A.2,*

$$-\frac{1}{\eta}\log\frac{\int_{B_r} f(\theta)\exp\{-\eta\{h(\theta) - h(\theta^*)\}\}d\theta}{\int_{B_r} f(\theta)d\theta} \leq rZ,$$

*where $B_r \triangleq \{\theta \in \Theta : \|\theta\| \leq r\}$.*

*Proof.* Note $\exp\{-\eta h(\theta)\} \geq \exp\{-\eta Z\|\theta\|\} \geq \exp\{-\eta rZ\}$ if $\theta \in B_r$. Thus,

$$-\frac{1}{\eta}\log\frac{\int_{B_r} f(\theta)\exp\{-\eta h(\theta)\}d\theta}{\int_{B_r} f(\theta)d\theta} \leq -\frac{1}{\eta}\log\frac{\int_{B_r} f(\theta)\exp\{-\eta rZ\}d\theta}{\int_{B_r} f(\theta)d\theta} = rZ.$$

$\square$

Back to the proof of Proposition 3.1. Suppose $\tilde{\theta}_j^*$ is any minimizer of $j$-th mode expected loss $\mathbb{E}_{(x,y)\sim\tilde{\mathcal{D}}_j}[L(M(x;\theta), y)]$. Let $\theta_t^* = \sum_{j=1}^m \tilde{\theta}_j^* \mathbb{1}(\mathcal{D}_t = \tilde{\mathcal{D}}_j)$, thus we have $\sum_{t=1}^{T-1}\mathbb{1}(\theta_{t+1}^* \neq \theta_t^*) = k_T$ and $\text{Card}(\{\theta_t^*\}_{t=1}^T) = m$. By Lemma A.2 with $h(\theta) = l_t(\theta)$ and taking expectation, for excess loss $e_t(\theta) \triangleq l_t(\theta) - l_t(\theta_t^*)$ we have,

$$\mathbb{E}\left[\int_\Theta f_{t-1}(\theta)e_t(\theta)d\theta\right] \leq -\frac{1}{\eta}\mathbb{E}\left[\log\int_\Theta f_{t-1}(\theta)\exp(-\eta e_t(\theta))\right] + \frac{\eta D^2 v^2}{2}. \tag{2}$$

Similarly by Lemma A.3, for any $r > 0$

$$-\frac{1}{\eta}\mathbb{E}\left[\log\frac{\int_{B_r(\theta_t^*)} f_{t-1}(\theta)\exp\{-\eta e_t(\theta)\}d\theta}{\int_{B_r(\theta_t^*)} f_{t-1}(\theta)d\theta}\right] \leq r\mathbb{E}[Z_t] \leq r\sqrt{\mathbb{E}[Z_t^2]} \leq rv., \tag{3}$$

where $B_r(\theta_t^*) \triangleq \{\theta \in \Theta : \|\theta - \theta_t^*\|_2 \le r\}$. Taking summation of inequalities equation 2 and equation 3, we have

$$
\begin{aligned}
\mathbb{E}\left[\mathrm{Regret}_T\right] =& \mathbb{E}\left[\sum_{t=1}^{T} \int_{\Theta} f_{t-1} l_t(\theta) - l_t(\theta_t^*)\right] \\
=& \mathbb{E}\left[\sum_{t=1}^{T} \int_{\Theta} f_{t-1} e_t(\theta)\right] \\
\text{(Inequality equation 2)} \le& -\frac{1}{\eta}\mathbb{E}\left[\sum_{t=1}^{T} \log \int_{\Theta} f_{t-1}(\theta)\exp(-\eta e_t(\theta))\right] + \frac{\eta D^2 v^2 T}{2} \\
\text{(Inequality equation 3)} \le& \frac{1}{\eta}\mathbb{E}\left[\sum_{t=1}^{T} \log \frac{\int_{B_r(\theta_t^*)} f_{t-1}(\theta)\exp\{-\eta e_t(\theta)\}d\theta / \int_{B_r(\theta_t^*)} f_{t-1}(\theta)d\theta}{\int_{\Theta} f_{t-1}(\theta)\exp(-\eta e_t(\theta))}\right] \\
& + \frac{\eta D^2 v^2 T}{2} + rvT.
\end{aligned}
$$
$$(4)$$

Since $\tilde{f}_t(\theta) = f_{t-1}(\theta)\exp\{-\eta l_t(\theta)\} / \int_{\Theta} f_{t-1}(\theta')\exp\{-\eta l_t(\theta')\}d\theta'$, we simplify the log term in the RHS as

$$
\log \frac{\int_{B_r(\theta_t^*)} f_{t-1}(\theta)\exp\{-\eta e_t(\theta)\}d\theta / \int_{B_r(\theta_t^*)} f_{t-1}(\theta)d\theta}{\int_{\Theta} f_{t-1}(\theta)\exp(-\eta e_t(\theta))} = \frac{\int_{B_r(\theta_t^*)} \tilde{f}_t(\theta)d\theta}{\int_{B_r(\theta_t^*)} f_{t-1}(\theta)d\theta}.
$$

Thus, inequality equation 4 is simplified as

$$
\mathbb{E}\left[\mathrm{Regret}_T\right] \le \frac{1}{\eta}\mathbb{E}\left[\sum_{t=1}^{T} \log \int_{B_r(\theta_t^*)} \tilde{f}_t(\theta)d\theta - \log \int_{B_r(\theta_t^*)} f_{t-1}(\theta)d\theta\right] + \frac{\eta D^2 v^2 T}{2} + rvT. \quad (5)
$$

It remains to connect $\tilde{f}$ and $f$. Note $f_{t-1} = (1 - \alpha_{t-1})\tilde{f}_{t-1} + \alpha_{t-1} g_{t-2}$, we have

$$
\begin{aligned}
-\log \int_{B_r(\theta_t^*)} f_{t-1}(\theta)d\theta \le& \left(-\log \int_{B_r(\theta_{t-1}^*)} \tilde{f}_{t-1}(\theta)d\theta + \log \frac{1}{1-\alpha_{t-1}}\right)\mathbb{1}(\theta_t^* = \theta_{t-1}^*) \\
& + \left(-\log \int_{B_r(\theta_t^*)} g_{t-2}(\theta)d\theta + \log \frac{1}{\alpha_{t-1}}\right)\mathbb{1}(\theta_t^* \ne \theta_{t-1}^*),
\end{aligned}
$$
$$(6)$$

Suppose $\theta_{n_i+1}^* = \cdots = \theta_{n_{i+1}}^*$ for $i = 0, \ldots, k-1$, where change points $0 = n_0 < n_1 < \cdots < n_k = T$. Assume $g_{t-2} = \sum_{j=0}^{t-2} c_{t-2,j} \tilde{f}_j$. For a segment $\theta_{n_i+1}^*, \ldots, \theta_{n_{i+1}}^*$, by the connection of $\tilde{f}$ and $f$ in inequality equation 6, we have

$$
\begin{aligned}
& \sum_{t=n_i+1}^{n_{i+1}} \left\{\log \int_{B_r(\theta_t^*)} \tilde{f}_t(\theta)d\theta - \log \int_{B_r(\theta_t^*)} f_{t-1}(\theta)d\theta\right\} \\
\le& \log \int_{B_r(\theta_{n_{i+1}}^*)} \tilde{f}_{n_i+1}(\theta)d\theta - \log \int_{B_r(\theta_{n_{i+1}}^*)} g_{n_i}(\theta)d\theta + \log \frac{1}{\alpha_{n_i}} \\
& + \sum_{t=n_i+2}^{n_{i+1}} \left\{\underbrace{\log \int_{B_r(\theta_t^*)} \tilde{f}_t(\theta)d\theta - \log \int_{B_r(\theta_t^*)} \tilde{f}_{t-1}(\theta)d\theta}_{\text{telescope}} + \log \frac{1}{1-\alpha_{t-1}}\right\} \\
=& \log \int_{B_r(\theta_{n_{i+1}}^*)} \tilde{f}_{n_{i+1}}(\theta)d\theta - \log \int_{B_r(\theta_{n_{i+1}}^*)} g_{n_i}(\theta)d\theta + \log \frac{1}{\alpha_{n_i}} + \sum_{t=n_i+1}^{n_{i+1}-1} \log \frac{1}{1-\alpha_t} \\
\le& \log \int_{B_r(\theta_{n_{i+1}}^*)} \tilde{f}_{n_{i+1}}(\theta)d\theta - \log \int_{B_r(\theta_{n_{i+1}}^*)} \tilde{f}_{n_{\phi(i)}}(\theta)d\theta + \log \frac{1}{c_{n_i,\phi(i)}\alpha_{n_i}} + \sum_{t=n_i+1}^{n_{i+1}-1} \log \frac{1}{1-\alpha_t},
\end{aligned}
$$
$$(7)$$

where $\phi(i) \triangleq \max\{j < i : \theta^*_{n_{j+1}} = \theta^*_{n_{i+1}}$ or $j = 0\}$ is the index of the endpoint of last segment with the same distribution. Combining inequality equation 5 and equation 7, we have

$$
\begin{aligned}
\mathbb{E}\left[\text{Regret}_T\right] \leq & \frac{1}{\eta} \sum_{j=1}^{m} \sum_{i:\mathcal{D}_{n_i+1}=\tilde{\mathcal{D}}_j} \sum_{t=n_i+1}^{n_{i+1}} \left\{ \log \int_{B_r(\tilde{\theta}^*_j)} \tilde{f}_t(\theta)d\theta - \log \int_{B_r(\tilde{\theta}^*_j)} f_{t-1}(\theta)d\theta \right\} \\
& + \frac{\eta D^2 v^2 T}{2} + rvT \\
\leq & \frac{1}{\eta} \sum_{j=1}^{m} \sum_{i:\mathcal{D}_{n_i+1}=\tilde{\mathcal{D}}_j} \left\{ \log \int_{B_r(\tilde{\theta}^*_j)} \tilde{f}_{n_{i+1}}(\theta)d\theta - \log \int_{B_r(\tilde{\theta}^*_j)} \tilde{f}_{n_{\phi(i)}}(\theta)d\theta - \log c_{n_i,\phi(i)} \right\} \\
& + \frac{\eta D^2 v^2 T}{2} + rvT + \frac{1}{\eta} C_{\text{Detect}} \\
\leq & \frac{1}{\eta} \sum_{j=1}^{m} \left\{ \log \int_{B_r(\tilde{\theta}^*_j)} \tilde{f}_{n_{\text{last},j}}(\theta)d\theta - \log \int_{B_r(\tilde{\theta}^*_j)} \tilde{f}_{n_0}(\theta)d\theta \right\} \\
& + \frac{\eta D^2 v^2 T}{2} + rvT + \frac{C_{\text{Detect}} + C_{\text{Remember}}}{\eta} \\
\leq & \frac{m}{\eta} \log \frac{\text{Vol}(\Theta)}{\text{Vol}(B_r(0))} + \frac{\eta D^2 v^2 T}{2} + rvT + \frac{C_{\text{Detect}} + C_{\text{Remember}}}{\eta} \\
\leq & \frac{md}{\eta} \log \frac{D}{2r} + \frac{\eta D^2 v^2 T}{2} + rvT + \frac{C_{\text{Detect}} + C_{\text{Remember}}}{\eta}
\end{aligned}
$$

where $n_{\text{last},j} \triangleq \max\{i : \theta^*_{n_i} = \tilde{\theta}^*_j\}$, $C_{\text{Detect}} \triangleq -\sum_{t=1}^{T-1} \log \tilde{\alpha}_t$, $\tilde{\alpha}_t \triangleq \alpha_t \mathbb{1}(s_{t+1} = s_t) + (1 - \alpha_t)\mathbb{1}(s_{t+1} \neq s_t)$, $\text{Vol}(A)$ is the volume for any set $A \subset \mathbb{R}^d$, and $C_{\text{Remember}} \triangleq -\sum_{i=1}^{k} \log c_{n_i,\phi(i)}$. Thus, by setting $r = md/(\eta v T)$, $\eta_{opt} = Dv\sqrt{T/\{2md \log(eDvT/2) + 2C_{\text{Remember}} + 2C_{\text{Detect}}\}}$ and note $\log x / x \leq 1$ for any $x > 0$,

$$
\begin{aligned}
\mathbb{E}\left[\text{Regret}_T\right] \leq & \frac{md}{\eta_{opt}} \log \frac{\eta_{opt} DvT}{2md} + \frac{\eta_{opt} D^2 v^2 T}{2} + \frac{md}{\eta_{opt}} + \frac{C_{\text{Detect}} + C_{\text{Remember}}}{\eta_{opt}} \\
\leq & \frac{md}{\eta_{opt}} \log \frac{DvT}{2} + \frac{\eta_{opt} D^2 v^2 T}{2} + \frac{md}{\eta_{opt}} + \frac{C_{\text{Detect}} + C_{\text{Remember}}}{\eta_{opt}} + 1 \\
= & Dv\sqrt{2T\left(md \log \frac{DvT}{2} + C_{\text{Detect}} + C_{\text{Remember}} + md\right)} + 1.
\end{aligned}
$$

In terms of cost functions,

$$
C_{\text{Detect}} = T\mathcal{H}(\frac{k}{T}) \leq k \log \frac{T}{k} + k \quad \text{if } \alpha_t \equiv \frac{k}{T},
$$

and

$$
C_{\text{Remember}} \leq \begin{cases} mT\mathcal{H}(\frac{k}{mT}) \leq k \log \frac{mT}{k} + k & \text{if } \gamma_t \equiv \frac{mT-k}{mT} \\ k \log T & \text{if } \gamma_t = \frac{1}{t}, \end{cases}
$$

where binary entropy $\mathcal{H}(p) \triangleq -p \log p - (1-p) \log(1-p)$.

$\square$

## A.2 PROOF OF PROPOSITION 3.2

*Proof.* By Proposition 3.1,

$$
\begin{aligned}
\sum_{t=1}^{T} \mathbb{E}\left[\int_{\Theta} f_{t-1} l_t(\theta) d\theta - l_t(\theta_t^*)\right] =& \mathbb{E}\left[\text{Regret}_T\right] \\
\leq& Dv\sqrt{2T\left(md\log\frac{eDvT}{2} + 2k\log\frac{T}{k} + k\log k + md\right)} + 1 \\
=& O(\sqrt{Tk\log T}) = o(T).
\end{aligned}
\tag{8}
$$

Since $\mathbb{E}[l_t(\theta)]$, $t \in [T]$, are uniformly strict, then for any $\varepsilon > 0$, there exists $\delta > 0$, such that

$$
\inf_{\theta:d(\theta_i,\mathcal{C}_t)\geq\varepsilon} \mathcal{E}[l_t(\theta)] - \min_{\theta'}\mathcal{E}[l_t(\theta')] \geq \delta,
$$

where $\mathcal{C}_t \triangleq \arg\min_{\theta\in\Theta} \mathbb{E}[l_t(\theta)]$ for all $t \in [T]$. By taking conditional expectations on history data on LHS,

$$
\begin{aligned}
\sum_{t=1}^{T} \mathbb{E}\left[\int_{\Theta} f_{t-1} l_t(\theta) d\theta - l_t(\theta_t^*)\right] =& \frac{1}{T}\sum_{t=1}^{T} \mathbb{E}\left[\mathbb{E}_{t-1}\left[\int_{\Theta} f_{t-1}\{l_t(\theta) - l_t(\theta_t^*)\}d\theta\right]\right] \\
=& \frac{1}{T}\sum_{t=1}^{T} \mathbb{E}\left[\int_{\Theta} f_{t-1}\{\mathbb{E}[l_t(\theta)] - \mathbb{E}[l_t(\theta_t^*)]\}d\theta\right] \\
\geq& \frac{\delta}{T}\sum_{t=1}^{T} \mathbb{E}\left[\int_{\theta\in\Theta:d(\theta,\cup_t\mathcal{C}_t)\geq\varepsilon} f_{t-1}(\theta)d\theta\right],
\end{aligned}
\tag{9}
$$

where $\mathbb{E}_{t-1}[\cdot]$ denotes the conditional expectation over the $\sigma$-algebra generated by $\{(x_\tau, y_\tau)\}_{\tau=1}^{t-1}$. Combining inequality equation 8 and equation 9, we have

$$
\frac{1}{T}\sum_{t=1}^{T} \mathbb{E}\left[\int_{\theta\in\Theta:d(\theta,\cup_t\mathcal{C}_t)\geq\varepsilon} f_{t-1}(\theta)d\theta\right] = o(1).
\tag{10}
$$

We connect $f$ with $\tilde{f}$ to get the convergence of knowledge $g_t$ as follows. Note $f_t = (1-\alpha_t)\tilde{f}_t + \alpha_{t-1}g_{t-1}$, then by equality equation 10 we have

$$
\begin{aligned}
\mathbb{E}\left[\int_{\theta\in\Theta:d(\theta,\cup_t\mathcal{C}_t)\geq\varepsilon} \frac{1}{T}\sum_{t=1}^{T} \tilde{f}_t(\theta)d\theta\right] \leq& \mathbb{E}\left[\int_{\theta\in\Theta:d(\theta,\cup_t\mathcal{C}_t)\geq\varepsilon} \frac{1}{T}\sum_{t=1}^{T} f_t(\theta)d\theta\right] \\
&+ \mathbb{E}\left[\int_{\theta\in\Theta:d(\theta,\cup_t\mathcal{C}_t)\geq\varepsilon} \frac{1}{T}\sum_{t=1}^{T} |f_t(\theta) - \tilde{f}_t(\theta)|d\theta\right] \\
\leq& o(1) + \frac{1}{T}\int_{\Theta}\sum_{t=1}^{T} \alpha_t \int_{\Theta} |g_{t-1} + \tilde{f}_t|d\theta \\
\leq& o(1) + 2\frac{1}{T}\sum_{t=1}^{T} \alpha_t = o(1).
\end{aligned}
\tag{11}
$$

The above result completes the first claim. For the second claim,

$$\mathbb{E}\left[\frac{1}{T}\sum_{t:\mathcal{D}_t=\tilde{\mathcal{D}}_j}\int_{\theta:d(\theta,\mathcal{B}_j)\leq\varepsilon}f_{t-1}(\theta)d\theta\right]$$

$$=\text{freq}_{j,T}-\mathbb{E}\left[\frac{1}{T}\sum_{t:\mathcal{D}_t=\tilde{\mathcal{D}}_j}\int_{\theta:d(\theta,\mathcal{B}_j)>\varepsilon}f_{t-1}(\theta)d\theta\right]$$

$$\geq\text{freq}_{j,T}-\mathbb{E}\left[\frac{1}{\delta T}\sum_{t:\mathcal{D}_t=\tilde{\mathcal{D}}_j}\int_{\theta:d(\theta,\mathcal{B}_j)>\varepsilon}f_{t-1}(\theta)\left\{\mathbb{E}[l_t(\theta)]-\min_{\theta'}\mathbb{E}[l_t(\theta')]\right\}d\theta\right]$$

$$\geq\text{freq}_{j,T}-\frac{1}{\delta}\mathbb{E}\left[\text{Regret}_T\right],$$

where $\mathcal{B}_j\triangleq\arg\min_{\theta\in\Theta}\mathbb{E}_{(x,y)\sim\tilde{\mathcal{D}}_j}\left[L(M(x;\theta),y)\right]$. Taking $\liminf_{T\to\infty}$ on both sides, we have

$$\liminf_{T\to\infty}\mathbb{E}\left[\frac{1}{T}\sum_{t:\mathcal{D}_t=\tilde{\mathcal{D}}_j}\int_{\theta:d(\theta,\mathcal{B}_j)\leq\varepsilon}f_{t-1}(\theta)d\theta\right]\geq\lim_{T\to\infty}\text{freq}_{j,T}-\lim_{T\to\infty}\frac{1}{\delta}\mathbb{E}\left[\text{Regret}_T\right]=q_t.$$

Since $\sum_{j=1}^m q_j=1$, for $\varepsilon>0$ that is small enough such that $\{\theta:d(\theta,\mathcal{B}_j)\leq\varepsilon\}$, $j\in[m]$, are disjoint, we have

$$1\geq\liminf_{T\to\infty}\mathbb{E}\left[\frac{1}{T}\sum_{j=1}^m\sum_{t:\mathcal{D}_t=\tilde{\mathcal{D}}_j}\int_{\theta:d(\theta,\mathcal{B}_j)\leq\varepsilon}f_{t-1}(\theta)d\theta\right]\geq\sum_{j=1}^m q_j=1$$

Combining these two inequalities, we have

$$\lim_{T\to\infty}\mathbb{E}\left[\frac{1}{T}\sum_{t=1}^T\int_{\theta:d(\theta,\mathcal{B}_j)\leq\varepsilon}f_{t-1}(\theta)d\theta\right]=q_j.$$

$\square$

# B  SCALABLE APPROXIMATION

## B.1  INTUITION

In Algorithm 1, we focus on two densities: the fast learner $\tilde{f}_t$ and the slow learner $g_t$. Assume there are $m_t$ Gaussian models, $\mathcal{N}(\beta_{t,i}, \sigma^2 I_d)$, constituting our knowledge base, and an additional Gaussian model, $\mathcal{N}(\theta_t, \sigma^2 I_d)$, for exploration. We approximate the fast and slow learner as the weighted averages of these $m_t + 1$ Gaussian models: ($\beta_{t,0} \triangleq \theta_t$)

$$\tilde{f}_t(\theta) \approx \sum_{i=0}^{m_t} w_{t,i} \mathcal{N}(\beta_{t,i}, \sigma^2 I_d), \quad g_t(\theta) \approx \sum_{i=0}^{m_t} r_{t,i} \mathcal{N}(\beta_{t,i}, \sigma^2 I_d),$$

where $\{w_{t,i}\}_{i=1}^{m_t}$ denote predictive mixing weights, $\{r_{t,i}\}_{i=1}^{m_t}$ represent slow weights, and $w_{t,0}$ and $r_{t,0}$ denote the cache weights to be consolidated. For simplicity, we only consider the first-order Taylor expansion of loss $l_t$, implying that the variance $\sigma^2$ remains unchanged. By substituting the approximation into the update rules in Algorithm 1, we obtain the weights update stated in Algorithm 2, and updates for $\theta_t$ and $\beta_{t,i}$ as:

$$\theta_{t+1} \leftarrow \theta_t - \eta \sigma^2 \nabla l_{t+1}(\theta_t), \quad \beta_{t+1,i} \leftarrow \beta_{t,i} - \eta \sigma^2 \frac{w_{t+1,i}}{\sum_{\tau=1}^{t+1} w_{\tau,i}} \nabla l_{t+1}(\beta_{t,i}).$$

In Algorithm 2, the fast learner is first updated using gradient descent (Line 5). After this update, predictive weights facilitate swift adaptation to prior tasks by recalling knowledge and exploring; this is achieved by mixing with slow weights and multiplying by their corresponding performances in Line 6. This process ensures that the agent remains responsive to new information.

In the context of knowledge consolidation, slow weights are first updated in Line 7. New information is selectively consolidated into the knowledge by applying gradient descent with varying step sizes (Line 7). These step sizes are determined by the relevance to the current data, namely the ratio of the predictive weight to the sum of historical predictive weights. This method enables the knowledge to refine by absorbing different amounts of current data while preventing forgetting.

In order to detect new modes, we monitor the patience, which is the sum of cache predictive weights $\max\{w_{t,0} - \tau, 0\}$ (Lines 8 to 10). When the patience surpasses a predetermined threshold $Q$, the current cache weight and fast learner are consolidated into the slow learner as a new component. This step ensures that the algorithm effectively responds to any new modes.

## B.2 DERIVATION

In this section, we will derive details of the scalable LEARN algorithm requiring only first-order derivatives. Let

$$
\begin{aligned}
\tilde{f}_t(\theta) &\approx w_{t,0}\mathcal{N}(\theta_t, \sigma^2 I_d) + \sum_{i=1}^{m_t} w_{t,i}\mathcal{N}(\beta_{t,i}, \sigma^2 I_d), \\
g_t(\theta) &\approx r_{t,0}\mathcal{N}(\theta_t, \sigma^2 I_d) + \sum_{i=1}^{m_t} r_{t,i}\mathcal{N}(\beta_{t,i}, \sigma^2 I_d).
\end{aligned}
\tag{12}
$$

Then the update rules in Line 5 and 6 of Algorithm 1 become:

$$
\begin{aligned}
\tilde{f}_{t+1}(\theta) \propto{}& f_t(\theta)\exp\{-\eta l_{t+1}(\theta)\} \\
\propto{}& \{(1-\alpha)w_{t,0} + \alpha r_{t,0}\}\exp\{-\eta l_{t+1}(\theta)\}\mathcal{N}(\theta_t, \sigma^2 I_d) \\
&+ \sum_{i=1}^{m_t}\{(1-\alpha)w_{t,i} + \alpha r_{t,i}\}\exp\{-\eta l_{t+1}(\theta)\}\mathcal{N}(\beta_{t,i}, \sigma^2 I_d) \\
\propto{}& w_{t+1,0}h_t(\theta_t)\exp\{-\eta g_{t+1}(\theta_t)^T(\theta-\theta_t)\}\exp\{-\frac{1}{2\sigma^2}\|\theta-\theta_t\|_2^2\} \\
&+ \sum_{i=1}^{m_t} w_{t+1,i}h_t(\beta_{t,i})\exp\{-\eta g_{t+1}(\beta_{t,i})^T(\theta-\beta_{t,i})\}\exp\{-\frac{1}{2\sigma^2}\|\theta-\beta_{t,i}\|_2^2\} \\
\propto{}& w_{t+1,0}h_t(\theta_t)\exp\left\{-\frac{1}{2\sigma^2}\|\theta-(\theta_t-\eta\sigma^2 g_{t+1}(\theta_t))\|_2^2\right\} \\
&+ \sum_{i=1}^{m_t} w_{t+1,i}h_t(\beta_{t,i})\exp\left\{-\frac{1}{2\sigma^2}\|\theta-(\beta_{t,i}-\eta\sigma^2 g_{t+1}(\beta_{t,i}))\|_2^2\right\} \\
\approx{}& w_{t+1,0}\mathcal{N}(\theta_{t+1}, \sigma^2 I_d) + \sum_{i=1}^{m_t} w_{t+1,i}\mathcal{N}(\beta_{t,i}-\eta\sigma^2 g_{t+1}(\beta_{t,i}), \sigma^2 I_d), \quad \text{(Omit } O(\eta^2))
\end{aligned}
\tag{13}
$$

where performance $h_t(\theta) \triangleq \exp\{-\eta l_{t+1}(\theta)\}$, gradient $g_{t+1}(\theta) \triangleq \nabla l_{t+1}(\theta)$, fast learner update

$$
\theta_{t+1} = \theta_t - \eta\sigma^2 g_{t+1}(\theta_t),
$$

predictive mixing weights

$$
w_{t+1,i} = c_{t+1}\left\{(1-\alpha)w_{t,i} + \alpha r_{t,i}\right\}h_t(\beta_{t,i}),
\tag{14}
$$

for $0 \le i \le m_{t-1}$ with $\beta_{t-1,0} \triangleq \theta_{t-1}$, and normalizer $c_{t+1}$ ensures $\sum_{i=1}^{m_t} w_{t+1,i} = 1$. The update of knowledge in Line 7 of Algorithm 1 is

$$
\begin{aligned}
g_{t+1}(\theta) ={}& \left\{\frac{tr_{t,0}}{t+1}\mathcal{N}(\theta_t, \sigma^2 I_d) + \frac{w_{t+1,0}}{t+1}\mathcal{N}(\theta_t - \eta g_{t+1}(\theta_t), \sigma^2 I_d)\right\} \\
&+ \sum_{i=1}^{m_t}\left\{\frac{tr_{t,i}}{t+1}\mathcal{N}(\beta_{t,i}, \sigma^2 I_d) + \frac{w_{t+1,i}}{t+1}\mathcal{N}(\beta_{t,i} - \eta g_{t+1}(\beta_{t,i}), \sigma^2 I_d)\right\} \\
\approx{}& r_{t+1,0}\mathcal{N}(\theta_{t+1}, \sigma^2 I_d) + \sum_{i=1}^{m_t} r_{t+1,i}\mathcal{N}(\beta_{t+1,i}, \sigma^2 I_d),
\end{aligned}
\tag{15}
$$

where

$$
\begin{aligned}
r_{t+1,i} &\triangleq r_{t,i} - \frac{r_{t,i} - w_{t,i}}{t} \\
\beta_{t+1,i} &\triangleq \beta_{t,i} - \eta\sigma^2\frac{w_{t+1,i}}{(t+1)r_{t+1,i}}g_{t+1}(\beta_{t,i}),
\end{aligned}
\tag{16}
$$

for $i \in [m_t]$.

*Remark* B.1. $r_{t,i} = \sum_{\tau=1}^{t} w_{\tau,i}/t$.

# C  IMPLEMENTATION OF LEARN

In traditional machine learning models, hyperparameters are typically tuned through a validation procedure. However, this is not possible for online data streams due to their continuous nature. To address this issue, in this section, we propose several practical strategies aimed at minimizing the number of hyperparameters that need tuning. These strategies will enhance the efficacy of our proposed algorithm in deep learning.

**Temperature and loss function**  Both the selection of temperature $\eta$ and loss function significantly impact the time required to learn a new pattern, as well as the model's robustness to noise. Higher temperature or an increased scale of loss function contributes to rapid convergence; however, it also harms the robustness against noise. To achieve scale-invariance, ensuring that scaling the loss function does not affect performance, we propose a solution: $\eta_t \triangleq \tilde{\eta}/l_t(\theta_{t-1})$ for a given fixed $\tilde{\eta}$. This way, the performance of LEARN remains consistent despite changes in the scale of the loss function.

**Refinement of knowledge**  In the refinement of the knowledge stage, each model in our knowledge set undergoes a gradient descent update with a learning rate proportional to $w_{t,i}/\sum_{\tau \le t} w_{\tau,i}$. In an effort to decrease computational load without sacrificing performance, we introduce an efficient strategy: only those models in the knowledge set with $w_{t,i} > \alpha$ are updated. We employ a truncated learning rate defined as

$$\beta_{t,i} \leftarrow \beta_{t-1,i} - \eta\sigma^2 \max\left\{\frac{w_{t,i}}{tr_{t,i}}, u_{\min}\right\}\nabla l_t(\beta_{t-1,i}),$$

where $u_{\min} \in (0,1)$ is a pre-specified value. We found that this approach reduces computation time and retains model performance effectively. However, it is important to note that alternative learning rate schedules could also be effective, and further exploration in this aspect is recommended.

**Choice of patience**  In our algorithm, we introduce a heuristic concept referred to as "patience". This heuristic is employed to monitor the predictive weight $w_{t,0}$ of the fast learner and manage its integration into our knowledge base. Specifically, once the patience level exceeds a predetermined threshold, we consider the fast learner ready for consolidation into the knowledge base. Practically speaking, this means a new model is created within the knowledge base when the sum of $\max\{0, w_{t,0} - \tau\}$, calculated from the last consolidation to the current step, surpasses a pre-set threshold, $Q$.

Aiming to reduce the number of hyperparameters needing tuning and accounting for potential stochastic variations in $w_{t,0}$, we propose a simple yet robust update rule for patience:

$$q_t \leftarrow q_{t-1} + \max\{-0.5\alpha, w_{t,0} + 2\alpha - 1\}.$$

This rule and the associated choices of hyperparameters are guided by the heuristic that, upon encountering a new data pattern, we expect the predictive weight $w_{t,0}$ to be approximately $1 - \alpha$.

**Pruning**  In practice, the knowledge base could contain redundant models. For example, it is possible for certain data distribution to occur only once but still be incorporated into the knowledge base, or for two models within the knowledge set to target the same data distribution. To mitigate such issues, we propose a straightforward pruning strategy for the knowledge base. This is made possible by leveraging additional information about the quality of each model, obtained through the knowledge weight $r_{t,i}$. A smaller knowledge weight indicates the limited usefulness of a model. Therefore, we opt to prune the model if its corresponding knowledge weight, $r_{t,i}$, falls below a certain threshold.

**Further improvement**  The scalable LEARN algorithm can be further improved by incorporating sophisticated strategies such as adaptive hyperparameters, guided by change-point detection techniques, and similarity measures. Change-point detection allows us to identify significant shifts in data sequences, informing the adjustment of the mixing parameter, $\alpha_t$, based on the belief of the occurrence of a change-point. Additionally, similarity measures between the fast learner and models in the knowledge set can guide the mixing ratio of a model in knowledge. The higher similarity suggests a higher mixing ratio, thereby enhancing efficiency.

## D   EXPERIMENTAL DETAILS

In this section, we detail the comparative methods and hyperparameters employed in our experiments. We also perform an ablation study, focusing on the selection of hyperparameters and the impact of the three core stages of our LEARN algorithm: exploration, recall, and refinement. Moreover, we expand the original Adaptive CL scenario via data augmentation, leading to a longer data stream, called Augmented Adaptive CL. Through two distinct sets of experiments conducted at different learning rates, our LEARN algorithm consistently outperforms in terms of average accuracy, knowledge accuracy, and adaptiveness.

### D.1   BASELINES

**Finetune**   The Finetune method operates by continuously updating a model using gradient descent on the current batch. This provides a naive baseline in Adaptive CL.

**Oracle**   The Oracle strategy maintains a set of distinct models, one for each task. During both training and inference, the task identity is given, and the corresponding model is employed for label prediction and current batch updates. This unrealistic approach serves as an upper bound for average accuracy, knowledge accuracy, and adaptiveness.

**ExpVAE**   ExpVAE keeps a set of tuples consisting of a classifier and generator, namely VAE, one tuple per task. During training, the task identity is known and the corresponding tuple is updated based on the current batch. In the inference phase, generators guide the selection of the classifier. The classifier with the minimum generator loss on the inputs is chosen. This approach is an upper bound of VAE-based inference methods.

**ER**   Experience Replay (ER)Chaudhry et al. [2019] presents a simple yet strong baseline in CL, without the need for known task identities. Utilizing Reservoir Sampling Vitter [1985], ER maintains a buffer where each data point is uniformly stored. During training, the model is updated on the integration of the current data batch and a batch sampled from this buffer.

**A-GEM**   Averaged Gradient Episodic Memory (A-GEM) Chaudhry et al. [2018a] is another method based on replay. Unlike ER, which integrates replayed data with current data, A-GEM calculates two separate gradients for the current and replayed data. By projecting the gradient over current data onto that of the replayed data, catastrophic forgetting can be effectively mitigated.

**Online EWC**   Online EWC [Chaudhry et al., 2018b] penalizes important parameters, where importance is the geometrically weighted average of the Fisher matrix.

**MIR**   MIR [Aljundi et al., 2019d] is a replay-based method that samples data in the buffer with the maximum scores.

**GDumb**   GDumb [Prabhu et al., 2020] propose a greedy sampler that greedily stores new samples so that classes are balanced. The model is only trained on memory, namely, batches sampled from the sampler.

We opt for ResNet18 He et al. [2016] with 64 initial filters for single-model approaches, namely Finetune, ER, A-GEM, Online EWC, MIR and GDumb. For mixture-model approaches, namely Oracle, ExpVAE, and LEARN, we use a reduced version of ResNet18 with 20 initial filters. We adopt a CNN-based VAE with two $3 \times 3$ convolutions in the encoders, the same as Lee et al. [2020].

In CL, comparing expansion-based and non-expansion methods directly may not lead to fair evaluations due to their inherent methodological differences. Expansion-based methods are typically more suited to handling dissimilar tasks or scenarios where negative transfer, a well-known issue in transfer learning Wang et al. [2019], is a concern. Non-expansion methods, on the other hand, may be advantageous for similar or related tasks. Despite these distinctions, merging elements from both methodologies could potentially enhance performance. For example, the knowledge base of LEARN can share certain architecture. Nevertheless, finding the optimal balance between these strategies is task-dependent and remains an open question.

Table 5: CIFAR10: comparison of different patience threshold $Q$ and mixing $\alpha$ from 3 runs. The selected hyperparameters in our experiment are denoted with a † with results from 10 runs.

| $(Q, \alpha)$ | Average Accuracy | Knowledge Accuracy | Adaptiveness |
|---|---|---|---|
| $(10, 0.2)$† | $72.70 \pm 0.07$ | $75.04 \pm 5.10$ | $71.85 \pm 0.08$ |
| $(10, 0.1)$ | $73.76 \pm 1.03$ | $78.38 \pm 4.34$ | $72.73 \pm 0.44$ |
| $(10, 0.3)$ | $71.34 \pm 0.18$ | $72.98 \pm 2.76$ | $70.18 \pm 0.24$ |
| $(10, 0.4)$ | $70.83 \pm 1.03$ | $71.32 \pm 2.44$ | $72.51 \pm 1.04$ |
| $(1, 0.2)$ | $73.35 \pm 1.00$ | $75.65 \pm 2.93$ | $72.13 \pm 0.41$ |
| $(5, 0.2)$ | $73.00 \pm 0.13$ | $77.73 \pm 2.18$ | $72.02 \pm 0.08$ |
| $(20, 0.2)$ | $71.85 \pm 0.26$ | $72.48 \pm 3.22$ | $70.15 \pm 0.28$ |

Table 6: CIFAR100: comparison of different patience threshold $Q$ and mixing $\alpha$ from 3 runs. The selected hyperparameters in our experiment are denoted with a † with results from 10 runs.

| $(Q, \alpha)$ | Average Accuracy | Knowledge Accuracy | Adaptiveness |
|---|---|---|---|
| $(10, 0.2)$† | $43.26 \pm 0.25$ | $41.22 \pm 2.02$ | $42.23 \pm 0.27$ |
| $(10, 0.1)$ | $41.23 \pm 0.34$ | $39.03 \pm 4.03$ | $40.20 \pm 0.45$ |
| $(10, 0.3)$ | $44.34 \pm 0.23$ | $42.59 \pm 2.36$ | $44.13 \pm 0.20$ |
| $(10, 0.4)$ | $40.83 \pm 0.40$ | $40.19 \pm 2.34$ | $39.41 \pm 0.34$ |
| $(1, 0.2)$ | $40.33 \pm 0.36$ | $40.58 \pm 2.83$ | $40.56 \pm 0.32$ |
| $(5, 0.2)$ | $44.02 \pm 0.43$ | $42.33 \pm 2.34$ | $43.30 \pm 0.33$ |
| $(20, 0.2)$ | $43.25 \pm 0.30$ | $41.44 \pm 2.09$ | $42.16 \pm 0.26$ |

### D.2 HYPERPARAMETERS

For the four Adaptive scenarios, we opt the same normalization with mean $0.5$ and standard deviation $0.25$. All methods employ the same learning rate of $0.001$. For LEARN, the patience threshold is set at $10$. In the pruning of knowledge, any model with a knowledge weight below $0.05$ is discarded. As a result, the maximum number of models in the knowledge base is limited $20$. For replay-based methods, namely ER and A-GEM, the maximum memory size is set to $1000$. ER utilizes a replay batch size of $5$, consistent with the current batch size, while A-GEM operates with a larger replay batch size of $64$.

### D.3 COMPUTATIONAL RESOURCES

All experiments conducted in this paper were carried out on a computational cluster equipped with 16 CPUs and powered by a Nvidia A100 GPU, having a total memory capacity of 50GB.

### D.4 REMARK ON ADAPTIVENESS

While the average accuracy implicitly indicates the ability to recall related knowledge encountering a new task, it is also affected by the ability to learn the current data pattern. Therefore, we propose a new *adaptiveness* metric, a segment-wise geometrically decaying weighted average of accuracy as follows: suppose the accuracy at time $t$ is $\text{acc}_t$, and change points of tasks are $0 = n_0 < n_1, \cdots < n_k = T$,

$$\text{Adaptiveness}_\gamma \left( \{\text{acc}_t\}_{t=1}^T \right) \triangleq \frac{\sum_{i=1}^k \sum_{\tau=1}^{n_{i+1}-n_i} \gamma^{\tau-1} \text{acc}_{n_i+\tau}}{\sum_{i=1}^k \sum_{\tau=1}^{n_{i+1}-n_i} \gamma^{\tau-1}} \in [0, 1].$$

Intuitively, a high value of Adaptiveness indicates a faster speed to recall related knowledge.

### D.5 ABLATION STUDY TABLES

We evaluate LEARN with different hyperparameters and conduct the ablation study over the three stages of LEARN, namely exploration, recall, and refinement.

Table 7: Mini-ImageNet: comparison of different patience threshold $Q$ and mixing $\alpha$ from 3 runs. The selected hyperparameters in our experiment are denoted with a † with results from 10 runs.

| $(Q, \alpha)$ | Average Accuracy | Knowledge Accuracy | Adaptiveness |
|---|---|---|---|
| $(10, 0.2)$† | $39.54 \pm 0.19$ | $36.08 \pm 4.98$ | $38.68 \pm 0.18$ |
| $(10, 0.1)$ | $40.00 \pm 0.19$ | $34.39 \pm 4.76$ | $38.97 \pm 0.18$ |
| $(10, 0.3)$ | $42.00 \pm 0.38$ | $38.01 \pm 5.43$ | $38.44 \pm 0.22$ |
| $(10, 0.4)$ | $37.36 \pm 0.28$ | $34.05 \pm 4.73$ | $36.63 \pm 0.23$ |
| $(1, 0.2)$ | $36.94 \pm 0.18$ | $33.64 \pm 4.92$ | $36.00 \pm 0.17$ |
| $(5, 0.2)$ | $39.77 \pm 0.19$ | $36.12 \pm 5.20$ | $38.69 \pm 0.34$ |
| $(20, 0.2)$ | $38.78 \pm 0.25$ | $35.52 \pm 4.91$ | $37.73 \pm 0.33$ |

Table 8: Tiny-ImageNet: comparison of different patience threshold $Q$ and mixing $\alpha$ from 3 runs. The selected hyperparameters in our experiment are denoted with a † with results from 10 runs.

| $(Q, \alpha)$ | Average Accuracy | Knowledge Accuracy | Adaptiveness |
|---|---|---|---|
| $(10, 0.2)$† | $34.57 \pm 0.12$ | $36.98 \pm 1.97$ | $32.82 \pm 0.15$ |
| $(10, 0.1)$ | $35.48 \pm 0.13$ | $37.91 \pm 2.79$ | $33.74 \pm 0.40$ |
| $(10, 0.3)$ | $34.21 \pm 0.34$ | $36.51 \pm 2.09$ | $32.49 \pm 0.27$ |
| $(10, 0.4)$ | $32.28 \pm 0.14$ | $34.83 \pm 2.00$ | $30.81 \pm 0.28$ |
| $(1, 0.2)$ | $36.12 \pm 0.21$ | $38.79 \pm 2.06$ | $34.43 \pm 0.18$ |
| $(5, 0.2)$ | $33.00 \pm 0.12$ | $34.97 \pm 1.90$ | $31.16 \pm 0.83$ |
| $(20, 0.2)$ | $35.61 \pm 0.12$ | $38.15 \pm 2.50$ | $33.71 \pm 0.17$ |

