# OpenReview forum: "Adaptive Continual Learning: Rapid Adaptation and Knowledge Refinement"
_ICLR.cc/2024/Conference — Submitted to ICLR 2024_

### Official Review · Reviewer_XjZn · 2023-10-28

**Soundness:** 2 fair
**Presentation:** 2 fair
**Contribution:** 2 fair
**Rating:** 5
**Confidence:** 4

**Summary:**

This paper introduces a new online continual learning setting in which there exists potential recurrence of tasks. It proposes an algorithm for this new settings called LEARN by exploiting the recurrence. It provides theoretical guarantees for the algorithm and offers a scalable implementation that leads to competitive empirical performance.

**Strengths:**

1. This paper introduces a new online continual learning setting where there is potential recurrence of tasks, and proposes a new algorithm for this new setting, which exploits the recurrence to improve the performance.
2. Theoretical guarantees are provided for the algorithm.

**Weaknesses:**

1. The paper is difficult to understand: it is not well organized and leaves out many details.

a) Figure 1 is too abstract and symbolic to be understood.

b) The authors refer to Figure 2 to illustrate the need for recall without further explanation.

c）Tempered Bayesian updates is a key component of the fast learner, but is not introduced.

d) There are some details missing from the proof. For example, the proof of Lemma A.2. applies Hoeffding's lemma directly, but the derivation is not straightforward, making it difficult to verify the correctness of the theory.

e) Important details, such as the derivation of Algorithm 2 and the definition of Adaptiveness are provided in Appendix, making it difficult to understand when reading the main text.



2. The discrepancy between the motivation and the algorithm.
As mentioned, "The primary goal is to activate the relevant slow learner for improved performance on seen tasks, and to utilize the fast learner for identifying and quickly learning new tasks."

However, there is no new task identification in the proposed algorithms. There is also no identification of relevant slow learners in Algorithm 1.
In Algorithm 1, the recall and refinement do not seem to leverage the knowledge of the recurrence of tasks. They are both updated by simply combining $\tilde{f}$ and $g$ with predefined ratios.

3. This paper does not discuss and compare the methods proposed for the very related setting where data of previous tasks/classes may appear again in CL, such as blurry task setting like "Koh et al. Online continual learning on class incremental blurry task configuration with anytime inference. In ICLR 2022", or the methods using fast and slow learner "Pham et al. DualNet: Continual Learning, Fast and Slow. In NeurIPS 2021"

4. Inappropriate choice of baselines in the experiments.
In Adaptive CL setting, there are no explicit task boundaries or identities. However, most baselines are not task-free methods. They are proposed under the assumption that there is an explicit task boundary, so they may not naturally work well in this setting. On the other hand, in Table 2, the only task-free baseline performs much better than the proposed method. The proposed method should be compared with more task-free baselines.

5. The randomness introduced by the random shuffling of 200 segments can have a significant impact on experimental performance.

**Questions:**

1. What are the definitions of Average Accuracy and Knowledge Accuracy? Can you provide their definition similar to the one of adaptiveness in Appendix D.4? In addition, why do the methods perform very differently in terms of knowledge accuracy and average accuracy?

2. Algorithm 2 needs to maintain 1+$m_{t-1}$ models. Why does it have almost the same number of trainable parameters as the other methods?

3. We do not know the number of distributions. Therefore we don't know $m_{t-1}$. How is $m_{t-1}$ obtained in Algorithm 2?

4. In the experiment, how many slower learners are there at the end of training?

---

> ### Author Response · Authors · 2023-11-17
> **Response 1: experimental response**
>
> Thank you for your comprehensive feedback and insightful suggestions. We appreciate your suggestions to improve clarity, and suggestions on experiments. In this rebuttal, we aim to address all your concerns comprehensively.
>
> ## Experiments
> ### Q: “compare … very related setting where data of previous tasks/classes may appear again in CL…”
> A: We appreciate the concerns. During the rebuttal stage, we add methods CLS-ER[3], CLIB [5], and DualNet [6]. In summary, these methods exibit less satisfactory performance in the results attached at the end. While this setting seems to be similar, we hope to emphasize that papers such as [1,2] focus on changing class distribution, where the key challenge is to handle possible class imbalance. However, our proposed scenario considers possibly recurring tasks, highlighting the need to rapidly recognize seen tasks and refine corresponding information.
>
> ### Q: “Inappropriate choice of baselines in the experiments.”
> A: To better clarify our choice of baseline methods, we want to mention that our selected methods (EWC++, MIR, ER, GDumb, AGEM) align with recent task-free research conventions [4,5,6,7]. These baselines are especially suitable for task-free scenarios because they do not require explicit task identity. For example, [7] used MIR, ER with GMED [8] variant; Similarly, the mentioned [4] used ER, MIR, GDUMB, and GMED, supplemented with methods designed for class imbalance – CBRS (Class-balancing reservoir sampling), ACE.
>
> ### Q: “What are Average Accuracy and Knowledge Accuracy…”
> “in Table 2, the only task-free baseline outperforms…”
>
> A: Thanks for the question. For better clarity, we have included the formulas for Average Accuracy and Knowledge Accuracy both in this rebuttal and the revised manuscript Average Accuracy measures the cumulative performance over time, and Knowledge Accuracy measures the information retained only at the last timestep, assuming task identity is known. Therefore, the method such as ExpVAE, which separately trains a classifier per task, has a high Knowledge Accuracy, while poor Average Accuracy since inference by VAEs is inaccurate.
> $
> \text{AverageAccuracy} = \frac{1}{T}\sum_{t=1}^T acc_t, \quad
> \text{Knowledge Accuracy} = \frac{1}{N} \sum_{i=1}^N acc_{T,i},$
> where $acc_t$ is the accuracy at time $t$, and $acc_{T,i}$ is the test accuracy of task $i$ assuming task identity is known at the final time.
>
> We want to emphasize that as indicated in Compared methods, Section 4.1, the baseline ExpVAE requires task identity during training, which separately trains a pair of generator and classifier for each task. In the inference stage, the generator is used to choose the correct classifier. ExpVAE serves as an upper bound on how such a structure of pairs of generators and classifiers will behave, even if task identities are known during training. As seen from the average accuracy in Table 1, inference from generator is inaccurate. Table 2 evaluates the final average test accuracy of the best classifier per task. Therefore the Knowledge accuracy of ExpVAE behaves similarly to Oracle which employs a multi-head ResNet with shared feature extractor, while during inference generators fail to correctly recall the correct classifier.
>
> ### Q: “random shuffling of 200 segments can have a significant impact on experimental performance.”
> A: Thanks for the insightful point. In order to avoid performance being significantly affected by randomness, we repeated experiments for 10 runs and reported the standard error. Since the standard error is moderate, we believe the reported results reflect the performance.
>
> ## Number of slow learners.
> ### Q: “Algorithm 2 needs to maintain $1+m_t$ models…almost the same number of trainable parameters as the other methods”
>
> A: We appreciate your observation. As mentioned in Architecture, Section 4.1, our proposed method and Oracle employ a reduced ResNet18 with a shared feature extractor. Therefore the excessive $m_t$ output heads contribute a small number of parameters compared with the feature extractor.
>
> ### Q: “How is $m_t$ obtained in Algorithm 2”
> A: Thanks for the excellent question. We want to clarify that slow learners are sequentially added once patience is above a threshold, and $m_t$ is the number of slow learners maintained at time $t$, which is not the underlying truth. To avoid confusion, we have revised the notation to $\hat{m}_t$ in the manuscript.
>
> ### Q: “How many slower learners are there at the end of training”
> A: Thanks for the helpful suggestion. Along with the maximum trainable parameters of our proposed algorithm reported in Table 4 (Section 4.1), the final count of slow learners at the end of training is detailed in Table 4, attached to the rebuttal.

---

> > ### Author Response · Authors · 2023-11-17
> > **Response 2: writing clarifications**
> >
> > ## Clarity
> > ### Q: “Figure 1 is too abstract”
> > A: Thanks for your feedback. We agree that a concrete algorithm illustration may better convey our intuition. Therefore, we have revised Figure 1 to provide a more detailed and concrete illustration of our algorithm, including its key components of one fast model, multiple slow models, and a router that keeps a mixing weight. Our algorithm is designed to let the router (1) assign a high weight to the relevant slow model when a prior task reoccurs, or to the fast model when a new task arrives; (2) update the corresponding model without information overwrite – namely one model already mastered one task continues to learn another task.
> >
> > ### Q: “Figure 2...without further explanation” “Tempered Bayesian updates … not introduced.”
> > A: Thanks for pointing out. To enhance clarity, we improve the explanations of Figure 2 in Section 3.1: “As illustrated in Figure 2, when the task switches to a previously learned task, a direct update without recall attempts to increase a small mass around the optimal parameter. However, once mixed with the slow density, that is, recall, the mass around the optimal parameter increases, resulting in faster recognition of the learned task.”
> >
> > We include a brief introduction of tempered Bayesian updates in Section 3.1. “The introduction of the temperature parameter $\eta$ is designed to moderate the impact of varying loss scales, especially since our approach does not rely on a probabilistic setting or the use of negative log-likelihood loss.”
> >
> > ### Q: “ proof of Lemma A.2. applies Hoeffding's lemma directly, but the derivation is not straightforward”  “ derivation of Algorithm 2 and the definition of Adaptiveness are provided in the Appendix, making it difficult to understand...”
> >
> > A: We are grateful for these suggestions. In the updated manuscript, we include the full derivation in the proof of Lemma A.2, and more details in other proofs in Appendix A. We also add key steps of the derivation of Algorithm 2 in Section 3.1, and definitions of three evaluation metrics: Average accuracy, Knowledge accuracy, and Adaptiveness in Metric, Section 4.1, to improve clarity.
> >
> > ## Discrepancy between the motivation and the algorithm
> > A: Thanks for the excellent question. We want to emphasize that Algorithm 1 implicitly recalls and identifies task identity, and scalable Algorithm 2 converts Algorithm 1 to a direct recall, identification, and refinement by Gaussian Mixture Model (GMM) approximation.
> > To better explain the connection between motivation and algorithm, we have moved key steps in the derivations in Appendix to Section 3.3. Intuitively, the slow density in Algorithm 1 is approximately a mixture density – a weighted sum of several components, each component represents information of one task. Here, (1) rapid task identification refers to a quick increase of mixing weight of corresponding task components; (2) recall refers to the process of fast density mixed with slow density – with such a mixing, it is easier to pull up the weight of the prior task component; (3) refinement refers to update of slow density.
> >
> >
> >
> >
> >
> >
> > [1] Koh et al. Online continual learning on class incremental blurry task configuration with anytime inference. In ICLR 2022
> >
> > [2] Koh H, Seo M, Bang J, et al. Online Boundary-Free Continual Learning by Scheduled Data Prior, ICLR. 2022.
> >
> > [3] Arani, Elahe, Fahad Sarfraz, and Bahram Zonooz. "Learning Fast, Learning Slow: A General Continual Learning Method based on Complementary Learning System." ICLR. 2021.
> >
> > [4] Simulating Task-Free Continual Learning Streams From Existing Datasets. CVPR23.
> >
> > [5] Koh et al. Online continual learning on class incremental blurry task configuration with anytime inference. In ICLR 2022
> >
> > [6] Pham et al. DualNet: Continual Learning, Fast and Slow. In NeurIPS 2021
> >
> > [7] Kumari, L., Wang, S., Zhou, T. and Bilmes, J.A., 2022. Retrospective adversarial replay for continual learning. Advances in Neural Information Processing Systems, 35, pp.28530-28544.
> >
> > [8] Jin, Xisen, et al. "Gradient-based editing of memory examples for online task-free continual learning." Advances in Neural Information Processing Systems 34 (2021): 29193-29205.

---

> > > ### Author Response · Authors · 2023-11-17
> > > **Supplementary: additional experimental results**
> > >
> > > As shown in Table 1 and 3, CLS-ER performs slightly worse than ER, likely due to its reliance on an Exponential Moving Average (EMA) model that may not adapt as swiftly to task switches. CLIB, designed for changing class distributions, maintains a memory buffer with sample importance, which is less applicable in our scenario where class imbalance is not a primary concern. While CLIB outperforms GDumb which also updates solely on memory, the average accuracy is not comparable to other methods. DualNet trains a fast model with features from prior tasks learned with semisupervised learning. From Table 1 and 3, we find the fast model cannot adapt to seen tasks as fast as our proposed light router structure. Table 2, focusing on knowledge accuracy, reveals that both CLS-ER and CLIB exhibit better information retention than Finetune, but not comparable to our proposed method and Oracle with multiple models.
> > > ## Tabel 1: Final average accuracy from 10 runs.
> > > |Method|CIFAR10|CIFAR100|Mini-ImageNet|Tiny-ImageNet|
> > > |--|--|--|--|--|
> > > |Oracle*|$77.38\pm0.08$|$45.50\pm0.13$|$41.68\pm0.10$|$36.37\pm0.07$|
> > > |ExpVAE*|$16.73\pm0.03$|$3.45\pm0.02$|$3.80\pm0.02$|$2.82\pm0.02$|
> > > |Finetune|$56.12\pm0.09$|$20.50\pm0.12$|$27.93\pm0.18$|$26.80\pm0.12$|
> > > |ER|$68.19\pm0.22$|$30.47\pm0.14$|$37.13\pm0.15$|$32.12\pm0.10$|
> > > |A-GEM|$64.16\pm0.11$|$22.40\pm0.07$|$29.81\pm0.14$|$28.40\pm0.35$|
> > > |Online EWC|$56.36\pm0.10$|$20.58\pm0.03$|$27.86\pm0.14$|$26.88\pm0.15$|
> > > |MIR|$59.22\pm0.40$|$19.39\pm0.18$|$29.41\pm0.22$|$28.28\pm0.12$|
> > > |GDumb|$36.69\pm0.24$|$8.74\pm0.06$|$11.50\pm0.03$|$5.22\pm0.07$|
> > > |**CLS-ER**|$65.17\pm0.27$|$28.16\pm0.33$|$35.66\pm0.15$|$30.85\pm0.16$|
> > > |**CLIB**|$38.33\pm0.87$|$11.51\pm0.47$|$13.42\pm0.24$|$6.74\pm0.24$|
> > > |**DualNet**|$61.56\pm0.13$|$26.50\pm0.12$|$32.62\pm0.19$|$30.08\pm0.33$|
> > > |LEARN|$\mathbf{72.70\pm0.07}$|$\mathbf{43.26\pm0.25}$|$ \mathbf{ 39.54\pm0.19}$|$\mathbf{34.57\pm0.12}$|
> > > ## Table 2: Knowledge accuracy from 10 runs
> > > |Method|CIFAR10|CIFAR100|Mini-ImageNet|Tiny-ImageNet|
> > > |--|--|--|--|--|
> > > |Oracle*|$81.91\pm2.64$|$51.92\pm1.14$|$50.85\pm1.43$|$43.08\pm1.11$|
> > > |ExpVAE*|$85.17\pm3.00$|$47.74\pm1.27$|$42.35\pm1.54$|$33.27\pm1.09$|
> > > |Finetune|$18.25\pm8.75$|$8.91\pm4.04$|$5.72\pm2.12$|$4.48\pm4.07$|
> > > |ER|$43.48\pm6.71$|$18.64\pm3.90$|$18.67\pm3.47$|$8.01\pm3.15$|
> > > |A-GEM|$25.45\pm8.11$|$10.83\pm4.55$|$6.90\pm2.43$|$4.71\pm3.96$|
> > > |Online EWC|$18.25\pm8.75$|$9.39\pm2.21$|$5.59\pm2.15$|$4.31\pm3.93$|
> > > |MIR|$18.49\pm8.84$|$10.39\pm3.85$|$7.80\pm2.38$|$5.36\pm3.92$|
> > > |GDumb|$37.71\pm4.15$|$9.66\pm1.01$|$12.66\pm0.98$|$5.42\pm0.53$|
> > > |**CLS-ER**|$51.38\pm9.48$|$23.24\pm4.83$|$25.26\pm3.79$|$14.69\pm3.67$|
> > > |**CLIB**|$42.74\pm5.12$|$19.77\pm1.09$|$22.24\pm1.43$|$10.40\pm0.57$|
> > > |**DualNet**|$24.75\pm7.05$|$13.19\pm4.41$|$10.73\pm2.65$|$12.80\pm4.44$|
> > > |LEARN|$\mathbf{75.04\pm5.10}$|$\mathbf{41.22\pm2.02} $|$\mathbf{36.08\pm4.98}$|$\mathbf{36.98\pm1.97}$|
> > > ## Table 3: Adaptiveness from 10 runs
> > > |Method|CIFAR10|CIFAR100|Mini-ImageNet|Tiny-ImageNet|
> > > |--|--|--|--|--|
> > > |Oracle*|$77.31\pm0.09$|$45.37\pm0.12$|$41.54\pm0.11$|$35.65\pm0.07$|
> > > |ExpVAE*|$16.67\pm0.02$|$3.37\pm0.02$|$3.92\pm0.02$|$2.92\pm0.03$|
> > > |Finetune|$53.32\pm0.09$|$18.37\pm0.10$|$25.74\pm0.17$|$21.36\pm0.10$|
> > > |ER|$67.00\pm0.23$|$28.55\pm0.13$|$35.96\pm0.14$|$28.66\pm0.10$|
> > > |A-GEM|$61.94\pm0.12$|$20.18\pm0.07$|$27.68\pm0.12$|$22.95\pm0.31$|
> > > |Online EWC|$53.57\pm0.10$|$18.42\pm0.03$|$25.70\pm0.13$|$21.40\pm0.14$|
> > > |MIR|$56.79\pm0.43$|$17.78\pm0.18$|$27.45\pm0.21$|$23.27\pm0.08$|
> > > |GDumb|$36.75\pm0.24$|$8.81\pm0.07$|$11.44\pm0.03$|$5.19\pm0.06$|
> > > |**CLS-ER**|$68.15\pm0.25$|$29.06\pm0.15$|$35.77\pm0.19$|$27.64\pm0.41$|
> > > |**CLIB**|$41.37\pm1.09$|$14.58\pm0.49$|$16.94\pm0.31$|$8.81\pm0.25$|
> > > |**DualNet**|$65.36\pm0.18$|$27.90\pm0.15$|$34.46\pm0.23$|$30.44\pm0.41$|
> > > |LEARN|$\mathbf{71.85\pm0.08} $|$\mathbf{42.23\pm0.27}$|$\mathbf{38.68\pm0.18}$|$\mathbf{32.82\pm0.15}$|
> > > ## Table 4: Number of slow learners (output heads) at the end (mean$\pm$standard error) from 10 runs.
> > > ||CIFAR10|CIFAR100|Mini-ImageNet|Tiny-ImageNet|
> > > |:--:|:-:|:--:|:--:|--|
> > > |Num of tasks|5|10|10|10|
> > > |Num of slow learners (output heads)|$5.50\pm0.22$|$11.10\pm0.35$|$11.00\pm0.37$|$11.70\pm0.58$|

---

### Official Review · Reviewer_heNc · 2023-10-30

**Soundness:** 3 good
**Presentation:** 3 good
**Contribution:** 3 good
**Rating:** 6
**Confidence:** 3

**Summary:**

In this paper, the authors introduce a new problem setting, Adaptive CL, considering recurring task environment without explicit task boundaries or identities. The authors then propose a LEARN algorithm including exploration, recall and refine process. Theoretical guarantees on online prediction with tight regret bounds and asymptotic consistency of knowledge are presented. Empirical evaluations are also done to show the effectiveness of the proposed LEARN algorithm.

**Strengths:**

Strength:

1.	A challenging new problem setting considering recurring task environment without explicit task boundaries or identities is presented.

2.	A LEARN algorithm for the Adaptive CL is proposed, and a scalable instantiation based on GMM is developed.

3.	Both theoretical and empirical analyses on LEARN are given.

**Weaknesses:**

Weakness:

Overall, the paper is well written, and the proposed new adaptive CL setting is practical and challenging. The proposed LEARN is technically sound and has been empirically verified to be effective in such a setting. Significant improvements over existing baselines are observed. The reviewer only has the following minor concerns.

1.	The method is verified on the classification task, as the problem setting is basically constructed according to label. The reviewer is wondering whether it is also applicable for regression problem. A related question is how to formulate the problem setting for regression task?

2.	The ablation studies can be further improved by considering each process only. From the current results, exploration plays the major role in achieving good results, and thus it is necessary to include exploration only results.

3.	For hyper-parameters Q and \alpha, why the current value range is selected? The appendix gives partial results on several combination of these two hyper-parameters, and it can be observed that the performance indeed varies with different settings. It is necessary to include a constructive guideline to set the two hyper-parameters, especially when facing new task or datasets.

**Questions:**

Please refer to the weakness.

---

> ### Author Response · Authors · 2023-11-17
>
> Thank you for your insightful comments. We are encouraged that you think the new scenario is practical and the proposed algorithm is technically sound and empirically verified. In the response, we aim to address the concerns adequately.
>
> ## Regression task
> Q: “whether it is also applicable for regression problem”
>
> A: Thanks for the insightful point. Although our experiments focused on classification tasks, a primary focus in computer vision, the proposed algorithm is applicable to any supervised learning problems including regression. In the regression case, the label is output which is usually a scalar and loss is typically the mean square error. The exact theoretical guarantees for the supervised case in Section 3 directly hold for such a scenario.
>
> ## Ablation study
> Q: “...considering each process only”
>
> A: Thanks for the helpful suggestion. We acknowledge that the exploration-only version serves as a useful baseline. In response to your query about considering each process only, we have updated the original Figure 5 in Section 4.2. From the figure, we find exploration-only version behaves worse than no-refinement and no-recall. Such a phenomenon is reasonable because while fast learner explores without memory the same as Finetune, slow learners never refine their knowledge and fast learner does not recall any knowledge from slow learners. Therefore, the exploration-only version quickly forgets learned tasks similar to Finetune.
>
> ## Choice of hyperparameters
> Q: “...constructive guideline to set the two hyper-parameters”
>
> A: Thanks for the excellent feedback. From the insights in theoretical guarantees, $\alpha$ is the proportion of slow learners mixed with fast learner, and $Q$ is the threshold to detect whether to create a new slow learner. Empirically, we find that setting $Q$ between $5$ and $20$, and $\alpha$ between $0.1$ and $0.2$, are good starting choices. For further optimization of the hyperparameters, conducting a grid search on the initial batches can yield more refined settings.
>
> Warm regards,
>
> Authors of Adaptive CL

---

### Official Review · Reviewer_TRLN · 2023-10-31

**Soundness:** 2 fair
**Presentation:** 2 fair
**Contribution:** 2 fair
**Rating:** 3
**Confidence:** 3

**Summary:**

This paper introduces a novel problem setting, Adaptive CL, inspired by human learning. It presents the LEARN algorithm, comprising three components: Exploration, Recall, and Refinement. The authors offer theoretical and experimental analysis to validate the effectiveness of the LEARN algorithm.

**Strengths:**

- The authors have developed a setting that more accurately mirrors human cognition and have provided a theoretical analysis to enhance understanding.

**Weaknesses:**

- The novelty of the constructed Adaptive CL setting is limited, since there are already some papers proposed similar periodic/recurring CL tasks [1,2].
- Some parts of the paper are hard to understand.
>-  In Figure 1, the meaning of the y-axis and the star symbol is unclear or this figure is just an illustration figure? It's also confusing why the third Refinement (blue lines) occurs before the second Recall (green arrows).
>- In Section 3.3, it’s unclear what the `scalability challenges' refer to. In Algorithm 2, the notation $\beta_{t-1,i}$ is also not explained.
>- At the time (t+1), it's not clear why the slow learners have the correct $m_t$ models, especially considering that Section 1.1 asserts that the method doesn't require knowledge of the task count. Is $m_t$ also indicative of the number of components in the GMM model? The paper needs to provide a more detailed explanation of how the GMM model is updated and how the relevant slow learner is selected in Figure 3.
- In the experimental section, the methods chosen for comparison are outdated. Considering that the proposed method incorporates the concept of the complementary system and emphasizes task-free CL, it should at least be compared with closely related methods such as Cls-ER[3] and recent task-free methods like [4]. Additionally, the paper should address the storage cost, as the method requires storing multiple slow learners.

[1] Koh H, Seo M, Bang J, et al. Online Boundary-Free Continual Learning by Scheduled Data Prior, ICLR. 2022.

[2] Xu, Zhenbo, Haimiao Hu, and Liu Liu. "Revealing the real-world applicable setting of online continual learning." 2022 IEEE 24th International Workshop on Multimedia Signal Processing (MMSP). IEEE, 2022.

[3] Arani, Elahe, Fahad Sarfraz, and Bahram Zonooz. "Learning Fast, Learning Slow: A General Continual Learning Method based on Complementary Learning System." ICLR. 2021.

[4] Simulating Task-Free Continual Learning Streams From Existing Datasets. CVPR23.

**Questions:**

See the three points in the Weaknesses.  As I still have questions regarding the experiments and found the paper difficult to understand, with some parts remaining unclear even after multiple readings, I recommend rejecting the paper in its current form.

---

> ### Author Response · Authors · 2023-11-17
> **Response 1**
>
> Thank you for your comprehensive and constructive feedback. We endeavor to address each of your concerns thoroughly in the subsequent response.
>
> ## Novelty
> ### Q: “...already some papers proposed similar periodic/recurring CL tasks [1,2]”
>
> A: Thanks for bringing up the relevant papers. Although these papers present scenarios that appear similar, the underlying challenges and corresponding solutions they address differ significantly from ours.
>
> These papers consider the scenario where class distributions change gradually or periodically over time. Thus the main challenge of this scenario is to learn without paying too much attention to the classes that frequently occur recently. Therefore, [1,2] focus on methods that can **balance class distribution**, such as an EMA (exponential moving average) of working models [1] and asymmetric cross entropy (ACE) to handle class imbalance [2].
>
> Our proposed Adaptive CL tackles a sequence of possibly recurring tasks without known task identities. The key challenge is to **quickly recall relevant information** when a prior task reoccurs, and accumulate knowledge over time. This challenge motivates us to develop a unique structure combining a single fast model, multiple slow models, and a router, specifically designed to rapidly recall relevant information for recurring tasks.
>
> ## Experiments
> ### Q:  “the methods chosen for comparison are outdated.” “compared with closely related methods such as Cls-ER[3] and recent task-free methods like [4]”
>
> A: Thanks for your excellent suggestions. During the rebuttal stage, we add methods CLS-ER[3], CLIB [5], and DualNet [6]. Since the focus of [4] is to propose a way to simulate a scenario with changing class distribution, [4] does not propose a new task-free algorithm. We show that these 3 new methods have less satisfactory performance. The results are discussed and attached at the end.
>
> To better clarify our choice of baseline methods, we want to mention that our selected methods (EWC++, MIR, ER, GDumb, AGEM) align with recent task-free research conventions [4,5,6,7]. These baselines are especially suitable for task-free scenarios because they do not require explicit task identity. For example, [7] used MIR, ER with GMED [8] variant; Similarly, the mentioned [4] used ER, MIR, GDUMB, and GMED, supplemented with methods designed for class imbalance – CBRS (Class-balancing reservoir sampling), ACE.
>
> ### Q: “address the storage cost”
>
> A: Thanks for your feedback. As indicated in Table 4, Section 4.1, the additional storage cost incurred by our algorithm is minimal.  Since we adopt a multi-head ResNet18 with a shared feature extractor, adding more heads introduces a small additional cost compared with the large shared feature extractor
>
>
> ## Clarity
> ### Q: “In Figure 1, the meaning … unclear”
>
> A1: Thanks for the helpful suggestion. To address this, we have revised Figure 1 in Section 1 to clearly depict our proposed network structure, showcasing the integration of fast/slow models and the router.
>
> ### Q: “scalability challenges refer to?”.
> A: Thanks for pointing out.  The term 'scalability challenges' in our context refers to the computational difficulties of managing probability distributions over a high-dimensional parameter space in previous algorithms. We have updated the expression in Section 3.3 to “However, this approach, which requires density over a high dimensional space, faces scalability challenges in large-scale deep learning tasks.”
>
> ### Q:  “In Algorithm 2, $\beta$ is also not explained.” “not clear why the slow learners have the correct $m_t$ models” “...more detailed explanation of how the GMM model is updated”
>
> Thanks for the excellent critic. Algorithm 2 involves three components at time $t$: (1) fast model $\theta_t$ (2) slow models $\beta_{t-1,i}$ for $i=1,\dots,m_{t-1}$ (3) mixing weights $w_{t,i}$ to combine fast/slow models.
> The $m_t$ is the estimated number of models in Algorithm 2. In Algorithm 2, when a fast learner consistently outperforms slow learners, it signals the emergence of a new task, prompting the addition of the fast learner as a new slow model.
> To improve clarity, in the updated manuscript, we will (1) move the key GMM approximation steps in the Appendix to Section 3.3, and (2) change $m_t$ to $\hat{m}_t$ to avoid confusion.

---

> ### Author Response · Authors · 2023-11-17
> **Supplementary: additional experimental results**
>
> As shown in Table 1 and 3, CLS-ER performs slightly worse than ER, likely due to its reliance on an EMA model that may not adapt as swiftly to task switches. CLIB, designed for changing class distributions, maintains a memory buffer with sample importance, which is less applicable in our scenario where class imbalance is not a primary concern. While CLIB outperforms GDumb which also updates solely on memory, the average accuracy is not comparable to other methods. DualNet trains a fast model with features from prior tasks learned with semisupervised learning. From Table 1 and 3, we find the fast model cannot adapt to seen tasks as fast as our proposed light router structure. Table 2, focusing on knowledge accuracy, reveals that both CLS-ER and CLIB exhibit better information retention than Finetune, but are not comparable to our proposed method and Oracle with multiple models.
> ## Tabel 1: Final average accuracy from 10 runs.
> |Method|CIFAR10|CIFAR100|Mini-ImageNet|Tiny-ImageNet|
> |-|-|-|-|-|
> |Oracle*|$77.38\pm0.08$|$45.50\pm0.13$|$41.68\pm0.10$|$36.37\pm0.07$|
> |ExpVAE*|$16.73\pm0.03$|$3.45\pm0.02$|$3.80\pm0.02$|$2.82\pm0.02$|
> |Finetune|$56.12\pm0.09$|$20.50\pm0.12$|$27.93\pm0.18$|$26.80\pm0.12$|
> |ER|$68.19\pm0.22$|$30.47\pm0.14$|$37.13\pm0.15$|$32.12\pm0.10$|
> |A-GEM|$64.16\pm0.11$|$22.40\pm0.07$|$29.81\pm0.14$|$28.40\pm0.35$|
> |Online EWC|$56.36\pm0.10$|$20.58\pm0.03$|$27.86\pm0.14$|$26.88\pm0.15$|
> |MIR|$59.22\pm0.40$|$19.39\pm0.18$|$29.41\pm0.22$|$28.28\pm0.12$|
> |GDumb|$36.69\pm0.24$|$8.74\pm0.06$|$11.50\pm0.03$|$5.22\pm0.07$|
> |**CLS-ER**|$65.17\pm0.27$|$28.16\pm0.33$|$35.66\pm0.15$|$30.85\pm0.16$|
> |**CLIB**|$38.33\pm0.87$|$11.51\pm0.47$|$13.42\pm0.24$|$6.74\pm0.24$|
> |**DualNet**|$61.56\pm0.13$|$26.50\pm0.12$|$32.62\pm0.19$|$30.08\pm0.33$|
> |LEARN|$\mathbf{72.70\pm0.07}$|$\mathbf{43.26\pm0.25}$|$ \mathbf{ 39.54\pm0.19}$|$\mathbf{34.57\pm0.12}$|
> ## Table 2: Knowledge accuracy from 10 runs
> |Method|CIFAR10|CIFAR100|Mini-ImageNet|Tiny-ImageNet|
> |-|-|-|-|-|
> |Oracle*|$81.91\pm2.64$|$51.92\pm1.14$|$50.85\pm1.43$|$43.08\pm1.11$|
> |ExpVAE*|$85.17\pm3.00$|$47.74\pm1.27$|$42.35\pm1.54$|$33.27\pm1.09$|
> |Finetune|$18.25\pm8.75$|$8.91\pm4.04$|$5.72\pm2.12$|$4.48\pm4.07$|
> |ER|$43.48\pm6.71$|$18.64\pm3.90$|$18.67\pm3.47$|$8.01\pm3.15$|
> |A-GEM|$25.45\pm8.11$|$10.83\pm4.55$|$6.90\pm2.43$|$4.71\pm3.96$|
> |Online EWC|$18.25\pm8.75$|$9.39\pm2.21$|$5.59\pm2.15$|$4.31\pm3.93$|
> |MIR|$18.49\pm8.84$|$10.39\pm3.85$|$7.80\pm2.38$|$5.36\pm3.92$|
> |GDumb|$37.71\pm4.15$|$9.66\pm1.01$|$12.66\pm0.98$|$5.42\pm0.53$|
> |**CLS-ER**|$51.38\pm9.48$|$23.24\pm4.83$|$25.26\pm3.79$|$14.69\pm3.67$|
> |**CLIB**|$42.74\pm5.12$|$19.77\pm1.09$|$22.24\pm1.43$|$10.40\pm0.57$|
> |**DualNet**|$24.75\pm7.05$|$13.19\pm4.41$|$10.73\pm2.65$|$12.80\pm4.44$|
> |LEARN|$\mathbf{75.04\pm5.10}$|$\mathbf{41.22\pm2.02} $|$\mathbf{36.08\pm4.98}$|$\mathbf{36.98\pm1.97}$|
> ## Table 3: Adaptiveness from 10 runs
> |Method|CIFAR10|CIFAR100|Mini-ImageNet|Tiny-ImageNet|
> |-|-|-|-|-|
> |Oracle*|$77.31\pm0.09$|$45.37\pm0.12$|$41.54\pm0.11$|$35.65\pm0.07$|
> |ExpVAE*|$16.67\pm0.02$|$3.37\pm0.02$|$3.92\pm0.02$|$2.92\pm0.03$|
> |Finetune|$53.32\pm0.09$|$18.37\pm0.10$|$25.74\pm0.17$|$21.36\pm0.10$|
> |ER|$67.00\pm0.23$|$28.55\pm0.13$|$35.96\pm0.14$|$28.66\pm0.10$|
> |A-GEM|$61.94\pm0.12$|$20.18\pm0.07$|$27.68\pm0.12$|$22.95\pm0.31$|
> |Online EWC|$53.57\pm0.10$|$18.42\pm0.03$|$25.70\pm0.13$|$21.40\pm0.14$|
> |MIR|$56.79\pm0.43$|$17.78\pm0.18$|$27.45\pm0.21$|$23.27\pm0.08$|
> |GDumb|$36.75\pm0.24$|$8.81\pm0.07$|$11.44\pm0.03$|$5.19\pm0.06$|
> |**CLS-ER**|$68.15\pm0.25$|$29.06\pm0.15$|$35.77\pm0.19$|$27.64\pm0.41$|
> |**CLIB**|$41.37\pm1.09$|$14.58\pm0.49$|$16.94\pm0.31$|$8.81\pm0.25$|
> |**DualNet**|$65.36\pm0.18$|$27.90\pm0.15$|$34.46\pm0.23$|$30.44\pm0.41$|
> |LEARN|$\mathbf{71.85\pm0.08} $|$\mathbf{42.23\pm0.27}$|$\mathbf{38.68\pm0.18}$|$\mathbf{32.82\pm0.15}$|
> ## Table 4: Number of slow learners (output heads) at the end from 10 runs.
> ||CIFAR10|CIFAR100|Mini-ImageNet|Tiny-ImageNet|
> |:-:|:-:|:-:|:-:|-|
> |Num of tasks|5|10|10|10|
> |Num of slow learners|$5.50\pm0.22$|$11.10\pm0.35$|$11.00\pm0.37$|$11.70\pm0.58$|
>
> [1] Koh H, Seo M, Bang J, et al. Online Boundary-Free Continual Learning by Scheduled Data Prior, ICLR, 2022.
>
> [2] Xu, Zhenbo, et al. Revealing the real-world applicable setting of online continual learning. MMSP, IEEE, 2022.
>
> [3] Arani, Elahe, et al. Learning Fast, Learning Slow: A General Continual Learning Method based on Complementary Learning System. In ICLR 2021.
>
> [4] Chrysakis, Aristotelis, et al. Simulating Task-Free Continual Learning Streams From Existing Datasets. In CVPR 2023.
>
> [5] Koh et al. Online continual learning on class incremental blurry task configuration with anytime inference. In ICLR 2022
>
> [6] Pham et al. DualNet: Continual Learning, Fast and Slow. In NeurIPS 2021
>
> [7] Kumari, Lilly, et al. Retrospective adversarial replay for continual learning. In NeurIPS 2022
>
> [8] Jin, Xisen, et al. Gradient-based editing of memory examples for online task-free continual learning.  In NeurIPS 2021

---

### Official Review · Reviewer_6Ke7 · 2023-10-31

**Soundness:** 2 fair
**Presentation:** 2 fair
**Contribution:** 2 fair
**Rating:** 5
**Confidence:** 2

**Summary:**

This paper studies an interesting topic, continual learning, which aims to learn a sequence of tasks without forgetting. The existing continual learning models are usually only considered to relieve forgetting in general continual learning. In contrast, this paper considers a new learning environment with possibly recurring tasks. This paper addresses this challenging setting by developing a new approach, achieving good results.

**Strengths:**

1. This paper is well-written.
2. The research topic in this paper is very interesting.

**Weaknesses:**

1. Some notations should be bold, such as x and y.
2. The actual network of the proposed model is not clear.
3. Why use the Regret Bound to explain the proposed approach instead of other theories?
4. The proposed approach requires three steps in each time, which leads to huge computational costs.

**Questions:**

Please see the weakness section.

---

> ### Author Response · Authors · 2023-11-17
>
> Thank you for your insightful suggestions and comments. We are pleased that you find the paper well-written and proposed Adaptive CL interesting. In this rebuttal, we aim to address all your concerns comprehensively.
>
> ##  Clarity.
> Q: “The actual network of the proposed model is not clear.”
>
> A: Thanks for your suggestions. We have updated Figure 1 to clearly illustrate the actual network architecture, comprising a fast model, multiple slow models, and a router that keeps a mixing weight to combine these models. Our algorithm guarantees that the router can (1) assign a high weight to the relevant slow model when a prior task reoccurs, or to the fast model when a new task arrives; (2) update the corresponding model without information overwrite, avoiding the case where a model already mastered one task continue to learn another and forget mastered one.
>
>
> ## Choice of Regret bound.
> Q: “Why use the Regret Bound to explain the proposed approach instead of other theories?”
>
> A: Thanks for mentioning the choice of regret instead of test accuracy in the end. In the online scenario, it's crucial to assess cumulative performance at each timestep, rather than just the final outcome. Therefore, regret, which measures the difference between the cumulative performance and the cumulative best performance when given the truth, is a standard metric in the literature of online learning.
>
>
> ## Computation cost.
> Q: “The proposed approach requires three steps in each time”
>
> A: Thanks for the excellent question. The router update, involving merely a one-dimensional vector, is computationally light. The major computations are the backpropagation of the fast model and several slow models. Similar to other methods involving multiple models [1,2], in the experiment, we keep the fast and slow models light – a multi-head ResNet18 with a shared feature extractor, so that the trainable parameters align with the method involving a single network. Given that each model is designed to handle a specific task, employing light models is both efficient and effective.
>
>
> [1] Lee, Soochan, et al. "A Neural Dirichlet Process Mixture Model for Task-Free Continual Learning." International Conference on Learning Representations. 2019.
>
> [2] Ye, Fei, and Adrian G. Bors. "Task-free continual learning via online discrepancy distance learning." Advances in Neural Information Processing Systems 35 (2022): 23675-23688.

---

### Author Response · Authors · 2023-11-17
**Global response**

Dear reviewers,

Thank you for your comprehensive reviews. Your feedback is invaluable in refining our work and ensuring its clear presentation. We are grateful for the positive remarks, particularly regarding the novel scenario and algorithm we proposed, and theoretical guarantees.

We've taken special note of the primary concerns in terms of clarity and experiments, and hope to summarize our clarifications in the following.

## Experiments.
### Additional experiments

Thanks for the insightful feedback from Reviewer heNc. We improve the ablation by including an exploration-only baseline, which reflects how the other two stages (refinement and recall) jointly contribute to a good performance.

Thanks for the suggestions from Reviewer TRLN and XjZn. We further evaluate methods from similar topics (CLS-ER [1], CLIB[2]), or similar dual structures (DualNet [3]). In summary, we find these methods are less satisfactory in our proposed setting. As shown in Table 1 and 3, CLS-ER performs slightly worse than ER, likely due to its reliance on an Exponential Moving Average (EMA) model that may not adapt as swiftly to task switches. CLIB, designed for changing class distributions, maintains a memory buffer with sample importance, which is less applicable in our scenario where class imbalance is not a primary concern. While CLIB outperforms GDumb which also updates solely on memory, the average accuracy is not comparable to other methods. DualNet trains a fast model with features from prior tasks learned with semisupervised learning. From Table 1 and 3, we find the fast model cannot adapt to seen tasks as fast as our proposed light router structure. Table 2, focusing on knowledge accuracy, reveals that both CLS-ER and CLIB exhibit better information retention than Finetune, but are not comparable to our proposed method and Oracle with multiple models.

### Choice of baselines
To clarify our choice of baseline methods, we want to mention that our selected methods (EWC++, MIR, ER, GDumb, AGEM) align with recent task-free research conventions [4,5,6,7]. These baselines are especially suitable for task-free scenarios because they do not require explicit task identity. For example, [7] used MIR, ER with GMED [8] variant; Similarly, the mentioned [4] used ER, MIR, GDUMB, and GMED, supplemented with methods designed for class imbalance – CBRS (Class-balancing reservoir sampling), ACE.


## Clarity
Thanks for Reviewer 6Ke7, TRLN, and XjZn. To address the concern that Figure 1 is hard to understand, in the updated manuscript, we revise Figure 1 in Section 1 with an illustration of our proposed algorithm structure, which consists of a fast model, multiple slow models, and a router that keeps a mixing weight to combine these models.

Warm regards,

Authors of Adaptive CL

[1] Arani, Elahe, et al. Learning Fast, Learning Slow: A General Continual Learning Method based on Complementary Learning System. In ICLR 2021.
.
[2] Koh H, Seo M, Bang J, et al. Online Boundary-Free Continual Learning by Scheduled Data Prior, ICLR, 2022.

[3] Pham et al. DualNet: Continual Learning, Fast and Slow. In NeurIPS 2021

[4] Chrysakis, Aristotelis, et al. Simulating Task-Free Continual Learning Streams From Existing Datasets. In CVPR 2023.

[5] Koh et al. Online continual learning on class incremental blurry task configuration with anytime inference. In ICLR 2022

[6] Pham et al. DualNet: Continual Learning, Fast and Slow. In NeurIPS 2021

[7] Kumari, Lilly, et al. Retrospective adversarial replay for continual learning. In NeurIPS 2022

---

> ### Author Response · Authors · 2023-11-17
> **Supplementary: additional experimental results**
>
> The following 4 tables report mean $\pm$ standard error from 10 runs.
> ## Tabel 1: Final average accuracy from 10 runs.
> |Method|CIFAR10|CIFAR100|Mini-ImageNet|Tiny-ImageNet|
> |--|--|--|--|--|
> |Oracle*|$77.38\pm0.08$|$45.50\pm0.13$|$41.68\pm0.10$|$36.37\pm0.07$|
> |ExpVAE*|$16.73\pm0.03$|$3.45\pm0.02$|$3.80\pm0.02$|$2.82\pm0.02$|
> |Finetune|$56.12\pm0.09$|$20.50\pm0.12$|$27.93\pm0.18$|$26.80\pm0.12$|
> |ER|$68.19\pm0.22$|$30.47\pm0.14$|$37.13\pm0.15$|$32.12\pm0.10$|
> |A-GEM|$64.16\pm0.11$|$22.40\pm0.07$|$29.81\pm0.14$|$28.40\pm0.35$|
> |Online EWC|$56.36\pm0.10$|$20.58\pm0.03$|$27.86\pm0.14$|$26.88\pm0.15$|
> |MIR|$59.22\pm0.40$|$19.39\pm0.18$|$29.41\pm0.22$|$28.28\pm0.12$|
> |GDumb|$36.69\pm0.24$|$8.74\pm0.06$|$11.50\pm0.03$|$5.22\pm0.07$|
> |**CLS-ER**|$65.17\pm0.27$|$28.16\pm0.33$|$35.66\pm0.15$|$30.85\pm0.16$|
> |**CLIB**|$38.33\pm0.87$|$11.51\pm0.47$|$13.42\pm0.24$|$6.74\pm0.24$|
> |**DualNet**|$61.56\pm0.13$|$26.50\pm0.12$|$32.62\pm0.19$|$30.08\pm0.33$|
> |LEARN|$\mathbf{72.70\pm0.07}$|$\mathbf{43.26\pm0.25}$|$ \mathbf{ 39.54\pm0.19}$|$\mathbf{34.57\pm0.12}$|
> ## Table 2: Knowledge accuracy from 10 runs
> |Method|CIFAR10|CIFAR100|Mini-ImageNet|Tiny-ImageNet|
> |--|--|--|--|--|
> |Oracle*|$81.91\pm2.64$|$51.92\pm1.14$|$50.85\pm1.43$|$43.08\pm1.11$|
> |ExpVAE*|$85.17\pm3.00$|$47.74\pm1.27$|$42.35\pm1.54$|$33.27\pm1.09$|
> |Finetune|$18.25\pm8.75$|$8.91\pm4.04$|$5.72\pm2.12$|$4.48\pm4.07$|
> |ER|$43.48\pm6.71$|$18.64\pm3.90$|$18.67\pm3.47$|$8.01\pm3.15$|
> |A-GEM|$25.45\pm8.11$|$10.83\pm4.55$|$6.90\pm2.43$|$4.71\pm3.96$|
> |Online EWC|$18.25\pm8.75$|$9.39\pm2.21$|$5.59\pm2.15$|$4.31\pm3.93$|
> |MIR|$18.49\pm8.84$|$10.39\pm3.85$|$7.80\pm2.38$|$5.36\pm3.92$|
> |GDumb|$37.71\pm4.15$|$9.66\pm1.01$|$12.66\pm0.98$|$5.42\pm0.53$|
> |**CLS-ER**|$51.38\pm9.48$|$23.24\pm4.83$|$25.26\pm3.79$|$14.69\pm3.67$|
> |**CLIB**|$42.74\pm5.12$|$19.77\pm1.09$|$22.24\pm1.43$|$10.40\pm0.57$|
> |**DualNet**|$24.75\pm7.05$|$13.19\pm4.41$|$10.73\pm2.65$|$12.80\pm4.44$|
> |LEARN|$\mathbf{75.04\pm5.10}$|$\mathbf{41.22\pm2.02} $|$\mathbf{36.08\pm4.98}$|$\mathbf{36.98\pm1.97}$|
> ## Table 3: Adaptiveness from 10 runs
> |Method|CIFAR10|CIFAR100|Mini-ImageNet|Tiny-ImageNet|
> |--|--|--|--|--|
> |Oracle*|$77.31\pm0.09$|$45.37\pm0.12$|$41.54\pm0.11$|$35.65\pm0.07$|
> |ExpVAE*|$16.67\pm0.02$|$3.37\pm0.02$|$3.92\pm0.02$|$2.92\pm0.03$|
> |Finetune|$53.32\pm0.09$|$18.37\pm0.10$|$25.74\pm0.17$|$21.36\pm0.10$|
> |ER|$67.00\pm0.23$|$28.55\pm0.13$|$35.96\pm0.14$|$28.66\pm0.10$|
> |A-GEM|$61.94\pm0.12$|$20.18\pm0.07$|$27.68\pm0.12$|$22.95\pm0.31$|
> |Online EWC|$53.57\pm0.10$|$18.42\pm0.03$|$25.70\pm0.13$|$21.40\pm0.14$|
> |MIR|$56.79\pm0.43$|$17.78\pm0.18$|$27.45\pm0.21$|$23.27\pm0.08$|
> |GDumb|$36.75\pm0.24$|$8.81\pm0.07$|$11.44\pm0.03$|$5.19\pm0.06$|
> |**CLS-ER**|$68.15\pm0.25$|$29.06\pm0.15$|$35.77\pm0.19$|$27.64\pm0.41$|
> |**CLIB**|$41.37\pm1.09$|$14.58\pm0.49$|$16.94\pm0.31$|$8.81\pm0.25$|
> |**DualNet**|$65.36\pm0.18$|$27.90\pm0.15$|$34.46\pm0.23$|$30.44\pm0.41$|
> |LEARN|$\mathbf{71.85\pm0.08} $|$\mathbf{42.23\pm0.27}$|$\mathbf{38.68\pm0.18}$|$\mathbf{32.82\pm0.15}$|
> ## Table 4: Number of slow learners (output heads) at the end from 10 runs.
> ||CIFAR10|CIFAR100|Mini-ImageNet|Tiny-ImageNet|
> |:--:|:-:|:--:|:--:|--|
> |Num of tasks|5|10|10|10|
> |Num of slow learners (output heads)|$5.50\pm0.22$|$11.10\pm0.35$|$11.00\pm0.37$|$11.70\pm0.58$|

---

### Meta-Review · Area_Chair_QEJj · 2023-12-06

**Metareview:**

(a) Summarize the scientific claims and findings of the paper based on your own reading and characterizations from the reviewers.
- Authors study a task-free continual-learning setting where tasks re-occur.
- The authors propose a method for learning (LEARN) under this setting.
- The authors demonstrate both theoretical properties (online regret bounds & ) of LEARN as well as empirical performance

(b) What are the strengths of the paper?
- The paper studies a challenging CL problem that seems to have practical use cases (task-free with task re-occurrence)
- The paper provides a theoretical analysis and a sound empirical study


(c) What are the weaknesses of the paper? What might be missing in the submission?
- The paper could be better positioned with respect to current literature (including related settings and baselines, even though the authors added some relevant ones in the rebuttal)
- [Minor] I was curious about the relationship between the proposed LEARN approach and meta-learning methods for continual learning. The ideas of slow and fast learners also seem similar, at least superficially, to meta-learning concepts. E.g., the setting in [1] has similarities (e.g., task recur, no task IDs).


[1]: Online Fast Adaptation and Knowledge Accumulation (OSAKA): a New Approach to Continual Learning, NeurIPS 2020

**Justification For Why Not Higher Score:**

This is more of a borderline paper than the provided ratings indicate. At the same time, there was no strong support from the reviewers for its acceptance. Several of the reviewers' criticisms relating to missing baselines, discussion of related literature, and some lack of clarity in the original manuscript were at least in part fixed by the authors through their revision.

Given the reviewers' comments, including in a brief private discussion, the consensus was that the paper still requires significant improvements in positioning vis-a-vis the current literature (including some of the baselines which you now compare to).

This is minor; the paper emphasizes the novelty of the setting, and contrasting it with other existing work would help. For example, there has been relevant work looking at recurrent tasks, including in settings that seem more challenging (e.g., see [1] above, where there are also concept drifts).

**Justification For Why Not Lower Score:**

N/A

---

### Decision · Program_Chairs · 2024-01-16

Reject